# Neural–Evolutionary Symbolic Regression with Global Constraints: Constraint-Aware Decoding and Reward Shaping

**Xiangdong Wu** [1]   **Wenjun Wu** [1 2]   **Ziyu Wei** [1]   **Bingrun Chen** [2]   **Zhenbo Song** [3]   **Rongye Shi** [1 2 *]

## Abstract

Symbolic regression aims to discover compact, interpretable mathematical expressions from data, but neural generation is challenging because expressions are tree-structured. Existing neural methods often linearize expression trees into token sequences, facilitating autoregressive modeling but obscuring hierarchical relations and complicating structure-dependent constraint enforcement. We propose **GCN-SR**, a graph-based symbolic regression framework that generates expressions in an explicit tree-aligned form, making structural context available during decoding. To enable batched generation over variable-topology expressions, we introduce Symbolic Perfect Binary Trees (SPBTs), a fixed-topology scaffold that preserves tree hierarchy while supporting graph-based node-attribute prediction. We further introduce Similarity-Weighted Policy Gradient (SWPG) to incorporate genetic programming (GP) refinement without directly imitating GP-refined elites; instead, refined expressions construct similarity-weighted rewards for samples drawn by the current generator. Experiments on standard symbolic regression benchmarks and ablations show that GCN-SR consistently improves exact recovery over strong neural and hybrid baselines under matched evaluation budgets.

## 1. Introduction

Symbolic regression (SR) aims to discover compact and interpretable mathematical expressions from observational data. Given a dataset $\mathcal{D} = \{(\mathbf{x}_n, y_n)\}_{n=1}^N$ with $\mathbf{x}_n \in \mathbb{R}^d$ and $y_n \in \mathbb{R}$, the goal is to recover a symbolic function $f : \mathbb{R}^d \to \mathbb{R}$ whose outputs match the observed responses. Unlike black-box regression, SR searches for the functional form itself, thereby offering mechanistic interpretability and potential extrapolation ability in scientific modeling (Brunton et al., 2016).

Symbolic expressions are naturally represented as trees, where internal nodes denote operators and leaves denote variables or constants. Classical genetic programming (GP) directly evolves such trees through mutation and crossover (Koza, 1994), but the search can become inefficient as the expression space grows. Recent neural SR methods instead learn proposal distributions over expressions, typically by linearizing expression trees into preorder token sequences and generating them with recurrent networks or Transformers (Petersen et al., 2021b; Kamienny et al., 2022; Holt et al., 2023b). This sequence formulation enables standard autoregressive modeling and efficient batching, and has led to strong empirical progress.

However, the sequence interface makes structure-dependent control less direct. Although a preorder sequence implicitly encodes an expression tree, key relations such as ancestor–descendant paths and subtree boundaries are not explicitly exposed. This matters because many useful SR constraints go beyond syntactic validity, aiming to avoid unstable or redundant forms such as repeated exp–log nesting, inverse-function cancellations, or deeply nested trigonometric compositions. Enforcing these constraints as hard rules requires partial tree context, especially the ancestors of the current node. While sequence decoders can recover this information through parser states, stacks, or rule-specific bookkeeping (Petersen et al., 2021b), they do not provide a unified explicit tree state for expressing general structural constraints as direct queries.

A natural alternative is to preserve the tree structure explicitly during generation. The difficulty is that expression trees have variable topology, which complicates vectorized decoding and neural message passing across a batch. To address this representation bottleneck, we introduce *Symbolic Perfect Binary Trees* (SPBTs), a fixed-topology scaffold that embeds variable-size symbolic expression trees into a shared tree structure by inserting placeholder nodes.

[1] School of Artificial Intelligence, Beihang University, China [2] Hangzhou International Innovation Institute, Beihang University, China [3] College of Computer Science and Engineering, Nanjing University of Science and Technology, China. Correspondence to: Rongye Shi <shirongye@buaa.edu.cn>.

*Proceedings of the 43rd International Conference on Machine Learning*, Seoul, South Korea. PMLR 306, 2026. Copyright 2026 by the author(s).

SPBT preserves the hierarchy of the expression tree while aligning node positions across samples, allowing symbolic generation to be formulated as node-attribute prediction on a fixed graph. This provides a unified structural interface for batched decoding, where ancestor-path and subtree-dependent constraints can be implemented through precomputed structural indices and masking, rather than through expression-specific parsing logic.

This tree-aligned interface further enables a graph neural generator to use the partially generated expression itself as the decoding state. At the same time, the fixed-topology scaffold introduces a trade-off: increasing the SPBT depth improves expressivity but causes the number of nodes to grow exponentially. Therefore, the neural generator is best viewed as a constrained proposal mechanism within a moderate-depth search space, rather than as a standalone exhaustive generator for arbitrarily deep formulas. To recover more complex expressions, we couple the neural generator with GP refinement. GP is well suited to explicit tree-space operations, since mutation and crossover can reorganize promising subexpressions from different sampled candidates into longer or higher-quality expressions.

However, directly feeding GP-refined elites back to the neural generator can create a distribution-mismatch problem. These refined expressions are produced by an evolutionary procedure rather than sampled from the current neural policy, and their sampling probabilities under the generator are generally unavailable. Directly imitating them may therefore make learning sensitive to off-policy elites. To mitigate this issue, we introduce *Similarity-Weighted Policy Gradient* (SWPG), which uses GP-refined elites only to construct similarity-weighted rewards for the original on-policy samples. In this way, GP discoveries can guide the generator without being treated as direct supervised targets. Together, the SPBT-based generator, constraint-aware decoding, GP refinement, and SWPG form our proposed framework, **GCN-SR**.

Our main contributions are as follows:

- We propose *Symbolic Perfect Binary Trees* (SPBTs), a fixed-topology scaffold that aligns variable-topology symbolic expressions while preserving explicit tree hierarchy.
- We develop a graph-based SPBT generator that casts expression generation as node-attribute prediction with structure-dependent constraint enforcement.
- We introduce *Similarity-Weighted Policy Gradient* (SWPG), a reward-shaping strategy that transfers feedback from GP-refined elites to structurally related on-policy samples.
- We integrate these components into **GCN-SR** and evaluate it on standard SR benchmarks, showing improved exact recovery under matched evaluation budgets.

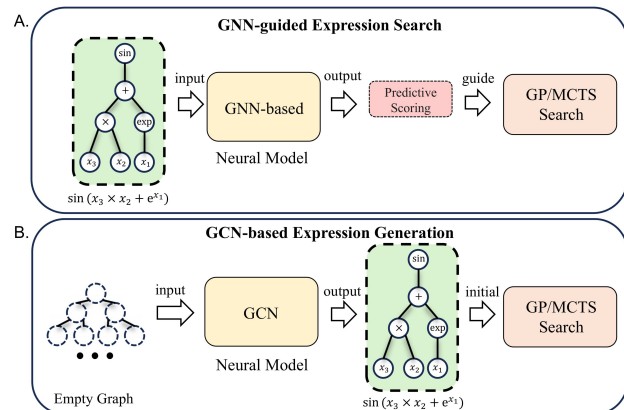

*Figure 1.* Two paradigms for using GNNs in symbolic regression. **A**. Prior methods mainly use GNNs as scoring or guidance models for external search procedures such as GP or MCTS. **B**. GCN-SR uses a GCN as the expression generator itself, decoding node attributes on a fixed SPBT scaffold with constraint-aware generation.

## 2. Related Work

Recent advances in symbolic regression increasingly incorporate learning-based models to improve search efficiency and scalability. Broadly, existing methods fall into three paradigms: (1) neural symbolic regression that generates expressions as token sequences; (2) hybrid neural–evolutionary frameworks that combine neural generators with GP-style search; and (3) GNN-based approaches that use graph representations to model expressions or guide the search. We briefly review these lines of work and position our method accordingly.

### 2.1. Neural-based Methods

Deep learning has been widely applied to symbolic regression to improve scalability and facilitate efficient exploration (Petersen et al., 2021b; Zhang et al., 2024; Biggio et al., 2021). Early systems such as AI Feynman (Udrescu & Tegmark, 2020b) combine neural components with physics-inspired heuristics and multi-stage pipelines to extract interpretable formulas. EQL (Kim et al., 2021) provides an end-to-end framework with a predefined operator library and sparsity constraints that encourage compact expressions, at the cost of a task-dependent inductive bias. DSR (Petersen et al., 2021b) casts SR as a sequential decision-making problem optimized with reinforcement learning, typically generating expressions in a token-by-token autoregressive manner. More recently, Transformer-based models such as E2ESR (Kamienny et al., 2022) and DGSR (Holt et al., 2023b) improve scalability and generalization in higher-dimensional settings.

Most neural SR methods linearize expression trees into token sequences, which enables standard sequence modeling

and batching but does not explicitly expose tree hierarchy during generation. As a result, enforcing *global* structure-dependent constraints (e.g., constraints on operator compositions along root-to-leaf paths) often requires additional structure recovery or bookkeeping, which is difficult to implement efficiently in batched decoding. In contrast, our method preserves an explicit tree-aligned representation during generation, enabling constraint-aware decoding while remaining batch-compatible.

## 2.2. Hybrid Neural–Evolutionary Methods

A growing body of work explores hybrid frameworks that integrate neural networks with genetic programming (GP) to combine complementary strengths: neural models provide fast, guided proposal distributions, while GP offers broad exploration through population-based variation. A common practice is to seed GP with neural-generated candidates and then evolve the population to obtain improved solutions. Methods such as GEGL (Ahn et al., 2020) and RSRM (Xu & Sun, 2024b) follow this decoupled pipeline, where the neural generator and GP are optimized independently.

More tightly coupled schemes attempt to use GP-refined elites to improve the neural generator. However, directly treating GP-refined expressions as training targets can be unstable when these elites lie far outside the generator's sampling distribution. NGGP (Mundhenk et al., 2021a) addresses this issue with Priority Queue Training (PQT), which maintains a buffer of high-reward expressions and updates the generator by maximizing their likelihood. While effective, such elite-matching objectives can overweight a small set of historical solutions and may reduce the diversity of the learned proposal distribution, depending on the buffer dynamics. In contrast, our method avoids directly learning from GP elites as targets; instead, it uses GP only to construct similarity-weighted reward signals for policy-sampled expressions, leveraging evolutionary refinement while mitigating distribution mismatch.

## 2.3. GNN-based Methods

GNNs have been used in SR pipelines in two main ways. First, they can serve as surrogate models of the input–output relation, after which a symbolic regressor approximates the surrogate with an interpretable expression (Cranmer et al., 2020; Reuter et al., 2023). Second, GNNs can be trained as scoring functions that guide search such as GP or Monte Carlo tree search (MCTS) (Wyrwiński & Krawiec, 2024; Xiang et al., 2025), as illustrated in Fig. 1A. In both cases, GNNs act mainly as *evaluators* rather than end-to-end *generators* of expressions.

Autoregressive graph generation has been extensively studied (Guo & Zhao, 2023; Liao et al., 2019; Brockschmidt et al., 2019; Chen et al., 2021; Zhong et al., 2022). Many approaches are trained in supervised regimes with explicit input–output pairs, and analogous supervised pipelines have been explored in SR to map data directly to expression trees (Zhong & Meidani, 2024; Kamienny et al., 2022). Such models typically optimize predictive fit, so the learned expressions can be numerically accurate yet structurally ambiguous, making exact recovery up to structural consistency or symbolic equivalence with the ground-truth formula challenging in many settings (Mundhenk et al., 2021a; Holt et al., 2023b; Xu & Sun, 2024b). Moreover, reinforcement learning for tree-structured generation brings practical hurdles: expressions must satisfy rigid syntactic and semantic constraints throughout decoding, while intermediate topologies vary across samples, hindering efficient batching.

Our method addresses these challenges by embedding expression trees into a *fixed-topology* scaffold, which reformulates symbolic tree generation as node attribute prediction over structurally aligned graphs. This design enables a GCN to serve as an end-to-end generator with an explicit structural inductive bias, while supporting constraint-aware, batch-compatible decoding (Fig. 1B).

## 3. Methods

GCN-SR integrates a graph-based neural generator with genetic programming (GP) in a *on-policy* training loop for symbolic regression. In each iteration, the generator samples a batch of candidate expressions, and GP performs local refinement initialized from these samples. Instead of treating GP outputs as training targets, we use them only to construct reward feedback for updating the generator via a Similarity-Weighted Policy Gradient (SWPG). We proceed in three steps: (i) we introduce a fixed-topology scaffold (SPBT) to enable batched tree generation; (ii) we leverage a GCN to autoregressively decode symbols on this scaffold; and (iii) we train the generator with SWPG, where GP influences learning only through reward construction.

### 3.1. The Framework of GCN-SR

The overall framework of GCN-SR is illustrated in Fig. 3B and summarized in Algorithm 1. GCN-SR consists of two components that interact in an iterative loop:

- **GCN-based neural generator.** A graph-based policy that samples candidate expressions by autoregressively assigning symbols on a fixed-topology scaffold, enabling batched decoding with explicit access to tree hierarchy.
- **Similarity-Weighted Policy Gradient (SWPG).** A learning scheme that runs GP refinement (a local evolutionary search) from the current batch, selects elites from the union of neural samples and GP-refined candidates, and assigns similarity-weighted rewards to the *original on-policy* samples for policy-gradient updates.

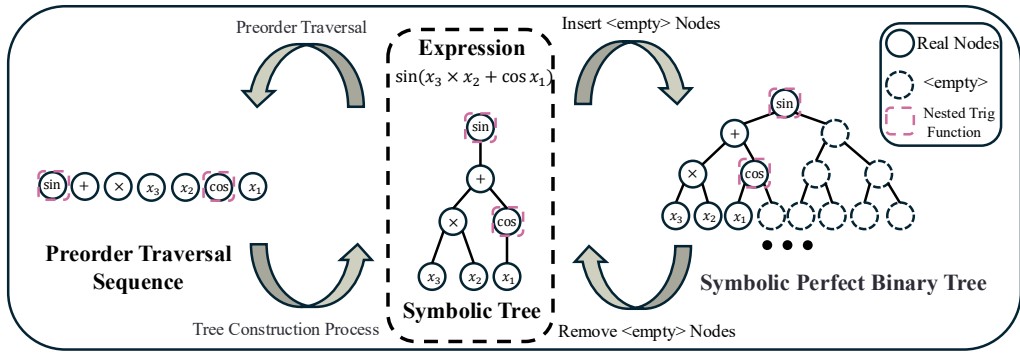

*Figure 2.* Two ways to represent the symbolic expression tree of $\sin(x_3 x_2 + \cos(x_1))$. **Left**: The preorder traversal sequence, which does not explicitly expose ancestor–descendant relations, making it difficult to enforce constraints defined over tree structure (e.g., disallowed operator patterns along a root-to-leaf path) from the sequence alone. **Right**: The same tree converted into the SPBT by adding `<empty>` nodes, which preserves explicit hierarchy and enables constraint checks via ancestor queries on the scaffold.

---

**Algorithm 1** GCN-SR (Overview)

---

**Require:** Dataset $\mathcal{D}$; symbol library $\mathcal{L}$; SPBT depth $D$; batch size $B$; elite size $K$; GP candidates $M$; max iterations $N_{\text{iter}}$; early-stop threshold $\varepsilon$
**Ensure:** Best discovered expression $\tau^\star$
 1: Initialize generator parameters $\theta$
 2: $\tau^\star \leftarrow \emptyset$;   $\text{MSE}^\star \leftarrow +\infty$
 3: **for** $t = 1$ **to** $N_{\text{iter}}$ **do**
 4:     Sample on-policy batch $\{\tau_\theta^b\}_{b=1}^B \sim \pi_\theta$ {Sec. 3.2}
 5:     Initialize GP with $\{\tau_\theta^b\}_{b=1}^B$ and obtain refined candidates $\{\tau^m\}_{m=1}^M$
 6:     $\mathcal{P} \leftarrow \{\tau_\theta^b\}_{b=1}^B \cup \{\tau^m\}_{m=1}^M$
 7:     $(\tau^\star, \text{MSE}^\star) \leftarrow \text{UPDATEBEST}(\mathcal{P}, \tau^\star, \text{MSE}^\star; \mathcal{D})$
 8:     **if** $\text{MSE}^\star \le \varepsilon$ **then**
 9:         **break**
10:     **end if**
11:     $\theta \leftarrow \text{SWPG}(\theta, \{\tau_\theta^b\}_{b=1}^B, \mathcal{P}, K; \mathcal{D})$ {Sec. 3.3}
12: **end for**
13: **return** $\tau^\star$

---

At iteration $t$, the generator samples an on-policy batch. GP refines these samples to produce additional candidates, forming a joint pool. SWPG selects elites from this pool and uses their fitness to construct reward signals for the on-policy samples; the generator is updated by policy gradient, without directly imitating GP-refined expressions.

### 3.2. GCN-based Neural Generator

Symbolic expressions are naturally represented as trees: internal nodes are operators (e.g., $\sin$, $+$) and leaves are variables or constants (e.g., $x$, $\pi$), as shown in Fig. 2. A major difficulty is that different expressions induce different topologies, which hinders vectorization and complicates batched, differentiable generation.

**SPBT: a fixed-topology scaffold.** To enable batch-compatible generation, we cast symbolic expression generation as node-attribute prediction on a shared fixed-topology tree. Since expressions in scientific modeling are typically composed of unary and binary operators (or can be binarized), we represent them as binary trees and normalize them into a **Symbolic Perfect Binary Tree** (SPBT): a perfect binary tree of depth $D$ (Fig. 2, right). Given an input tree of depth at most $D$, we insert a placeholder token `<empty>` for each missing child and apply this padding recursively until all internal nodes have two children and all leaves lie at depth $D$. This converts variable-shaped expression trees into a fixed scaffold, allowing generation to be treated as node-attribute prediction on an aligned graph.

This normalization is reversible: removing all `<empty>` nodes recovers the original valid expression tree. Moreover, once a consistent child ordering is fixed, a symbolic expression is uniquely determined by the symbols assigned to SPBT nodes. Thus, each expression corresponds to one node-attribute assignment on the fixed SPBT topology.

**GCN modeling on SPBT-SL.** With a fixed graph structure, symbolic regression can be viewed as node-attribute prediction on a fixed tree-structured graph (cf. graph attribute completion (Jin et al., 2021)). We adopt a Graph Convolutional Network (GCN) to model attribute dependencies on this graph, and progressively assign symbols to SPBT nodes. To preserve each node's identity during message passing, we augment SPBT with self-loops for GCN computation, forming SPBT-SL. The symmetrically normalized adjacency matrix $\hat{A}$ is:

$$\hat{A} = \tilde{D}^{-1/2}(A + I)\tilde{D}^{-1/2}, \qquad (1)$$

where $A$ encodes parent–child edges, $I$ represents self-loops, and $\tilde{D}$ is the degree matrix of $A + I$.

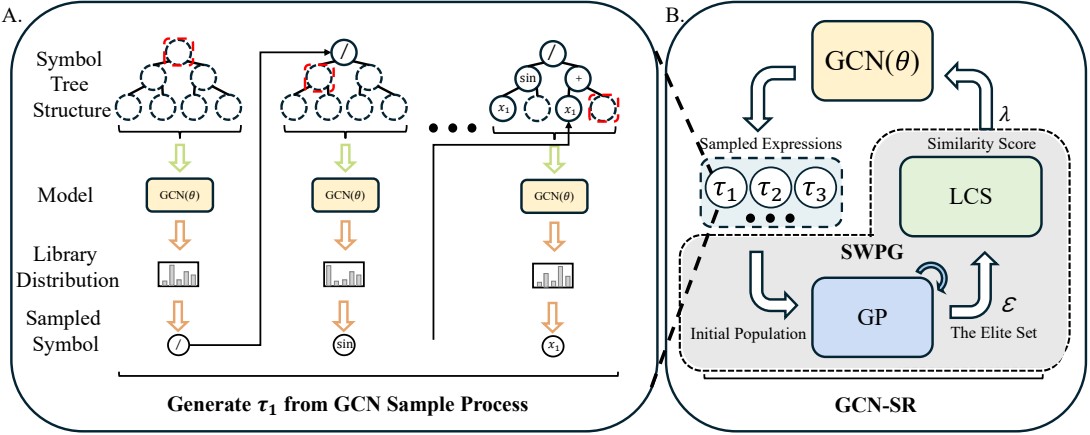

*Figure 3.* **A.** Expression sampling based on GCNs, leveraging matrix operations to encode global structural information. The red dashed box marks the current node being sampled. **B.** The GCN-SR overview. SWPG integrates GP and GCN in a strictly on-policy manner to mitigate distribution mismatch between the neural policy and GP-refined expressions.

**Autoregressive decoding.** Based on SPBT-SL, we generate expressions by autoregressively assigning symbols to SPBT nodes in preorder. At step $i$, we maintain a feature matrix $X_i \in \mathbb{R}^{|\mathcal{V}| \times (|\mathcal{L}|+1)}$, where each row is the one-hot encoding of the symbol currently assigned to a node (or `<empty>` if unassigned). We run an $L$-layer GCN on the fixed scaffold to obtain node representations:

$$H_i^{(0)} = X_i W^{(0)}, \qquad (2)$$

$$H_i^{(\ell+1)} = \sigma\left(\hat{A} H_i^{(\ell)} W^{(\ell)}\right), \quad \ell = 0, \ldots, L-1, \quad (3)$$

where $W^{(\ell)}$ are trainable weights and $\sigma$ is an activation function. The representation of the current node $v_i$ is $h_i = H_i^{(L)}[i]$. We sample the next symbol from a masked categorical distribution $\pi_i = \text{softmax}(W_o^\top h_i)$, where invalid symbols are masked out according to the applicable constraints. We then update $X_{i+1}$ by replacing the $i$-th row with the one-hot encoding of the sampled symbol. Each decoding step performs one GCN forward pass on the fixed scaffold; thus the sampling cost scales with $|\mathcal{V}| = 2^D - 1$ and the GCN depth $L$.

This repeats until all nodes are processed, yielding a completed SPBT. The final expression is obtained by taking the preorder traversal and removing all `<empty>` tokens, resulting in a sequence $\tau = [a_0, a_1, \ldots, a_{T-1}]$ with $T \le |\mathcal{V}|$.

**Constraint-aware decoding.** To ensure validity and numerical stability during sampling, we enforce three types of constraints. Many of these rules depend on ancestor–descendant structure (e.g., restricting operator compositions along a root-to-leaf path). SPBT keeps this hierarchy explicit and consistently indexed across the batch, enabling constraint-aware decoding via batched masking.

Concretely, at each step we build a boolean validity mask over $\mathcal{L}$ for each sample and apply it to the logits before sampling. Masks are computed from (i) the already-assigned node labels and (ii) their ancestor context on the fixed scaffold (depth $D$). With a fixed node ordering, constraint checks can be implemented as batched tensor gathers using a precomputed ancestor table, rather than reconstructing per-sample partial trees or maintaining dynamic stacks as in sequence decoding.

- **Structural constraints:** We fix the maximum tree depth to $D$, which limits the SPBT size to $|\mathcal{V}| = 2^D - 1$. This is analogous to a maximum sequence length in DSR (Petersen et al., 2021b), while using tree depth as a natural inductive bias for hierarchical expressions.
- **Syntactic constraints:** (i) unary operators must have their right child as `<empty>`; (ii) leaf nodes must be assigned a variable or constant; (iii) if a node is `<empty>`, all its descendants are implicitly `<empty>` and need not be sampled, pruning irrelevant subtrees.
- **Semantic constraints:** We forbid: (1) trigonometric compositions (e.g., $\sin(x + \cos(x))$); (2) invalid inverse-cancellation and overflow-prone exp–log nesting (e.g., $\exp(\log(x))$; $\exp(x + \exp(x))$); (3) trigonometric terms inside logarithms (e.g., $\log(\sin(x)+\cos(x))$), which helps reduce redundant identities, prevent frequent domain violations, and improve numerical stability in evaluation.

When constants are included in $\mathcal{L}$, their numeric values are treated as free parameters and optimized post-hoc for each sampled structure using nonlinear solvers (e.g., BFGS) to maximize reward (Petersen et al., 2021b). This inner optimization does not backpropagate through the generator; it only provides a reward for the sampled expression structure.

The sampling procedure is illustrated in Fig. 3A.

### 3.3. Similarity-Weighted Policy Gradient

Having established a graph-based neural generator, we now describe how to leverage GP refinement to guide learning without directly training on GP-refined expressions as targets. We consider the standard policy objective $J_{\text{std}}(\theta) = \mathbb{E}_{\tau \sim \pi_\theta}[R(\tau)]$, where $R(\tau)$ is the data-fitting reward (e.g., $R(\tau) = \frac{1}{1+\text{MSE}(\tau;\mathcal{D})}$).

A naive way to incorporate GP-refined expressions is to treat them as additional samples in the score-function update, leading to the following heuristic mixed gradient:

$$\nabla \hat{J}_{\text{mix}}(\theta) = \frac{1}{B+M} \left( \sum_{b=1}^{B} A(\tau_\theta^b) \nabla_\theta \log \pi_\theta(\tau_\theta^b) \right.$$
$$\left. + \sum_{m=1}^{M} A(\tau^m) \nabla_\theta \log \pi_\theta(\tau^m) \right),$$
$$(4)$$

where $A(\tau) = R(\tau) - \bar{R}$ is an empirical advantage with $\bar{R}$ computed on the same mixed batch, $\{\tau_\theta^b\}_{b=1}^{B} \sim \pi_\theta$ are on-policy samples, and $\{\tau^m\}_{m=1}^{M}$ are GP-refined off-policy samples. However, this direct mixing is generally biased and can be unstable under distribution mismatch: GP may concentrate on regions that are poorly covered by $\pi_\theta$, so a small number of off-policy terms can dominate the update and amplify variance. A natural way to avoid these issues is to update the policy using only on-policy samples. The remaining question is how to use high-quality GP discoveries to *shape* learning while keeping the score-function term strictly on-policy.

**SWPG: reward shaping via structural similarity.** In our framework, GP is initialized from on-policy samples and applies *local* structural edits. As a result, GP-refined expressions are typically close to their originating samples (e.g., sharing subtrees). When a refined expression attains higher fitness, this provides a credit signal for the shared structure. SWPG leverages this by constructing a similarity-shaped reward for each on-policy sample, while computing gradients *only* through on-policy log-probabilities.

Concretely, SWPG draws an on-policy batch $\{\tau_\theta^b\}_{b=1}^{B} \sim \pi_\theta$, runs GP refinement to generate additional candidates, and selects an elite set $\mathcal{E} = \{\tau^{(1)}, \ldots, \tau^{(K)}\}$ from the union of neural samples and GP variants. For each on-policy sample $\tau_\theta^b$, we define the similarity-weighted reward

$$r_b = r(\tau_\theta^b, \mathcal{E}) = \frac{1}{K} \sum_{k=1}^{K} \lambda(\tau_\theta^b, \tau^{(k)}) R(\tau^{(k)}), \quad (5)$$

where $\lambda(\tau_\theta^b, \tau^{(k)}) \in [0, 1]$ measures structural similarity. Within a policy update, we treat the elite set $\mathcal{E}$ as *fixed* (stop-

gradient), and optimize the following similarity-shaped surrogate objective: $J_{\text{sim}}(\theta; \mathcal{E}) = \mathbb{E}_{\tau \sim \pi_\theta}[r(\tau, \mathcal{E})]$. Importantly, the gradient is taken only through $\pi_\theta$; GP affects the update solely via the fixed set $\mathcal{E}$.

**Proposition 3.1** (SWPG gradient estimator (stop-gradient surrogate)). *Fix an elite set $\mathcal{E}$ during a policy update and consider $J_{\text{sim}}(\theta; \mathcal{E})$ above. Let $\{\tau_\theta^b\}_{b=1}^{B} \sim \pi_\theta$ be i.i.d. samples and $r_b = r(\tau_\theta^b, \mathcal{E})$. Then an on-policy Monte Carlo estimator of $\nabla_\theta J_{\text{sim}}(\theta; \mathcal{E})$ is*

$$\nabla_\theta \hat{J}_{\text{sim}}(\theta) = \frac{1}{B} \sum_{b=1}^{B} (r_b - \bar{r}) \nabla_\theta \log \pi_\theta(\tau_\theta^b), \quad (6)$$

*where $\bar{r} = \frac{1}{B} \sum_{b=1}^{B} r_b$, is a plug-in baseline used for variance reduction.*

We next show that this estimator is bounded and its variance decreases with the batch size.

**Proposition 3.2** (Boundedness and variance (stop-gradient surrogate)). *Assume: (i) $0 < R(\tau) \leq 1$ for any candidate $\tau$; (ii) $0 \leq \lambda(\tau, \tau^{(k)}) \leq 1$ for all $\tau$ and elites $\tau^{(k)} \in \mathcal{E}$; (iii) $\|\nabla_\theta \log \pi_\theta(\tau)\|_2 \leq C$ for some $C > 0$ and all $\tau$; (iv) $\mathcal{E}$ is fixed during the update. Then: (i) $\|\nabla_\theta \hat{J}_{\text{sim}}(\theta)\|_2 \leq C$; (ii) $\text{Var}\left(\nabla_\theta \hat{J}_{\text{sim}}(\theta)\right) \leq \frac{4C^2}{B}$.*

Together, Propositions 3.1–3.2 show that SWPG provides a well-behaved *on-policy* gradient estimator for the stop-gradient surrogate objective $J_{\text{sim}}(\theta; \mathcal{E})$, with controlled magnitude and variance scaling as $O(1/B)$ (Appendix B).

**Similarity function.** To compute $\lambda(\cdot, \cdot)$ efficiently, we compare expressions via their preorder token sequences. Given an on-policy sample $\tau_\theta^b$ and an elite $\tau^{(k)}$, we define $\lambda(\tau_\theta^b, \tau^{(k)})$ as the Longest Common Subsequence (LCS) length between the two preorder sequences, normalized by $|\tau_\theta^b|$. With a fixed child ordering, any shared subtree induces an identical token subsequence in preorder, so the LCS provides a simple proxy for structural overlap. Before computing LCS, we apply lightweight canonicalization to reduce superficial representational differences, including algebraic simplification and deterministic reordering for commutative operators; for example, we map $x_2 + x_1$ to the canonical form $x_1 + x_2$.

The similarity score is:

$$\lambda(\tau_\theta^b, \tau^{(k)}) = \frac{\text{LCS}(\tau_\theta^b, \tau^{(k)})}{|\tau_\theta^b|}, \quad (7)$$

where normalization by $|\tau_\theta^b|$ yields an asymmetric measure of how much of the on-policy sample is retained in the elite.

In summary, SWPG integrates local evolutionary refinement into a strictly on-policy learning framework by using GP

exclusively for reward construction. This preserves stable score-function updates while allowing the generator to exploit structural regularities discovered by GP search.

## 4. Experiments and Results

### 4.1. Evaluation Metrics

We evaluate performance using the *recovery rate* (Petersen et al., 2021b), defined as the fraction of independent runs that exactly recover the ground-truth expression. To account for randomness and data scale, we compute recovery rates across different sample sizes $N$ over $J$ independent runs, where each run uses a newly sampled dataset.

Formally, let $\mathcal{D}_j^N = \{(x_n, f(x_n))\}_{n=1}^N$ denote the dataset of size $N$ sampled from the ground-truth function $f$ in the $j$-th run. Given an algorithm $\mathcal{A}$, the recovery rate is:

$$\text{Recovery Rate}(N) = \frac{1}{J} \sum_{j=1}^J \mathbb{I}\left[\mathcal{A}(\mathcal{D}_j^N; \mathcal{L}) \equiv f\right], \quad (8)$$

where $\mathcal{L}$ is the shared symbol library and $\equiv$ denotes symbolic equivalence. Unless stated otherwise, methods are compared under a matched evaluation budget measured by the number of evaluated candidate expressions per run (counted whenever a candidate is scored), terminating upon recovery or budget exhaustion.

### 4.2. Baselines and Benchmarks

In our comparison, we include the following SR baselines: **GP**, standard genetic programming; **DSR**, a policy-gradient-based sequence generator (Petersen et al., 2021b); **NGGP**, a hybrid method that uses an RNN generator with GP refinement and updates the generator using GP-improved candidates (Mundhenk et al., 2021a); **DGSR**, a Transformer-based symbolic regression model (Holt et al., 2023b); and **RSRM**, which combines double Q-learning with Monte Carlo tree search (Xu & Sun, 2024b).

We evaluate all methods on three widely used benchmarks at three data scales ($N \in \{20, 100, 1000\}$), using multiple random datasets per $N$: **Nguyen**, a standard suite of analytic expressions with one or two variables; **R\*** (rational functions), which includes three rational expressions with polynomial terms in both numerator and denominator; and **Livermore**, a challenging suite featuring exponentials, trigonometric terms, and highly nonlinear polynomials.

### 4.3. Main Performance Results

As shown in Table 1, GCN-SR achieves the best recovery rates across all three benchmarks and all evaluated data scales. Notably, GCN-SR remains competitive even in the low-sample regime ($N = 20$), indicating strong data efficiency. Across $N \in \{20, 100, 1000\}$, recovery rates are broadly consistent, with improvements on some tasks as more data become available. In addition, the standard deviations are generally small, suggesting stable behavior across repeated runs with different random datasets.

Further implementation details and computational analysis are provided in Appendix C.5.1– C.5.5, including wall-clock statistics, constant-handling experiments, and representative recovered expressions (with MSE and complexity) on tasks where exact symbolic recovery is not achieved.

### 4.4. Ablation Study

We conduct ablation studies on the Livermore benchmark to isolate the contributions of key components in GCN-SR. All variants use the same symbol library $\mathcal{L}$, training budget (total number of fitness evaluations), and GP configuration; only the specified component is modified. Table 3 reports exact recovery rates (%) over 10 independent runs.

**Model 1 (w/o Semantic Constraints)** removes the semantic-constraint set (i.e., the global semantic checks) while keeping the depth limit and syntactic rules unchanged. **Model 2 (w/o GP)** disables GP refinement and relies solely on the GCN generator. **Model 3 (w/o Reward Refinement)** removes similarity-based reward shaping and updates the generator using the raw data-fitting reward of each on-policy sample, while keeping GP as a decoupled post-processor. **Model 4 (w/o Similarity Score $\lambda$)** sets $\lambda = 1$ in SWPG, so all on-policy samples receive the same elite-averaged reward; this ablates structure-aware credit assignment. **Model 5 (w/ PQT)** replaces SWPG with Priority Queue Training (PQT) (Mundhenk et al., 2021a) while keeping the GCN generator, SPBT representation, constraints, and GP refinement unchanged. **Model 6 (Random GCN)** freezes the generator to sample symbols uniformly at random (no learning), while keeping the downstream GP refinement and evaluation budget unchanged.

*Table 3.* Ablation study on the Livermore tasks. Recovery rates (%) are averaged over 10 independent runs.

| Model Variant | Recovery Rate (%) |
|---|---|
| Full GCN-SR | $89.5 \pm 28.0$ |
| **w/o Semantic Constraints** | $86.4 \pm 29.5$ |
| **w/o GP** | $29.5 \pm 38.7$ |
| **w/o Reward Refinement** | $80.5 \pm 37.1$ |
| **w/o Similarity Score $\lambda$** | $76.8 \pm 37.8$ |
| **w/ PQT** | $82.3 \pm 33.8$ |
| **Random GCN** | $58.2 \pm 46.0$ |

Table 3 shows that GP refinement is a major contributor to recovery: removing GP (Model 2) causes a large drop, indicating that local evolutionary search substantially improves over pure neural sampling. Among training schemes, removing similarity-based reward refinement (Model 3) or

*Table 1.* Recovery rate (%) of several algorithms on the Nguyen, Livermore, and R* benchmark problem sets with different data counts. Results are computed at the task level: for each task, we estimate recovery as the fraction of successes over 20 independent runs. We then report the mean recovery rate across tasks, with uncertainties given by the sample standard deviation across tasks.

| Benchmark | Data Count | GP | DSR | DGSR | NGGP | RSRM | GCN-SR |
|---|---|---|---|---|---|---|---|
| Nguyen | 20 | 61.7% ± 45.9% | 84.2% ± 32.3% | 79.2% ± 39.6% | 91.7% ± 28.9% | 83.3% ± 38.9% | **93.3% ± 23.1%** |
| | 100 | 62.5% ± 44.1% | 85.8% ± 30.6% | 81.7% ± 36.9% | 91.7% ± 28.9% | 85.0% ± 35.3% | **93.3% ± 23.1%** |
| | 1000 | 62.5% ± 46.5% | 87.5% ± 31.1% | 82.5% ± 36.7% | 91.7% ± 28.9% | 92.5% ± 23.0% | **93.3% ± 23.1%** |
| Livermore | 20 | 17.3% ± 25.5% | 30.5% ± 38.7% | 77.3% ± 36.1% | 77.7% ± 37.4% | 78.6% ± 34.0% | **90.0% ± 26.7%** |
| | 100 | 23.2% ± 27.1% | 29.1% ± 42.1% | 78.6% ± 32.7% | 76.4% ± 34.7% | 83.2% ± 35.4% | **90.5% ± 26.5%** |
| | 1000 | 19.5% ± 26.1% | 30.0% ± 40.8% | 78.2% ± 34.5% | 79.5% ± 35.0% | 85.5% ± 31.1% | **91.8% ± 26.7%** |
| R* | 20 | 0.0% ± 0.0% | 0.0% ± 0.0% | 10.0% ± 17.3% | 13.3% ± 23.1% | 96.7% ± 5.8% | **100.0% ± 0.0%** |
| | 100 | 0.0% ± 0.0% | 0.0% ± 0.0% | 40.0% ± 36.1% | 33.3% ± 57.7% | **100.0% ± 0.0%** | **100.0% ± 0.0%** |
| | 1000 | 0.0% ± 0.0% | 13.3% ± 23.1% | 46.7% ± 41.6% | 33.3% ± 57.7% | **100.0% ± 0.0%** | **100.0% ± 0.0%** |

*Table 2.* Results on the 100 Feynman equations under 0.1% relative noise. We report the exact symbolic recovery rate for all methods.

| GPlearn | GP-GOMEA | AFP | AFP-FE | DSR | uDSR | AI Feynman | PySR | GCN-SR |
|---|---|---|---|---|---|---|---|---|
| 18/100 | 23/100 | 24/100 | 26/100 | 24/100 | 40/100 | 38/100 | 53/100 | 61/100 |

structure-aware similarity weighting (Model 4) decreases recovery on average, suggesting that SWPG and $\lambda$ provide useful credit assignment beyond using elites alone. Replacing SWPG with PQT (Model 5) also reduces performance, consistent with the benefit of strictly on-policy reward shaping in our setting. Finally, the Random GCN baseline (Model 6) still achieves non-trivial recovery due to downstream GP refinement, but remains far below the full model, highlighting the advantage of a learned structure-aware generator. Additional comparisons (including direct GCN vs. sequence-based generators under matched training) are provided in the Appendix C.5.1.

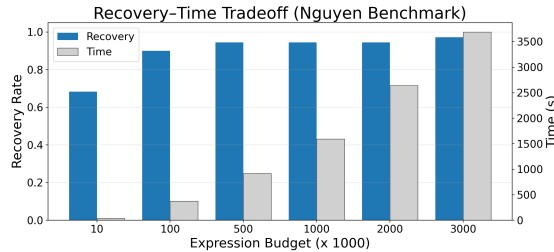

*Figure 4.* Tradeoff Experiments

### 4.5. SRBENCH

We further evaluate GCN-SR on SRBENCH (La Cava et al., 2021), an open-source benchmarking suite that provides a standardized pipeline over diverse regression tasks and many state-of-the-art symbolic regression baselines. It includes the Feynman Equations suite (white-box) with known ground-truth formulas (Feynman et al., 2015), as well as black-box tasks where the target forms are unknown. Following the official protocol, we report the benchmark *solution rate*, i.e., the fraction of equations whose recovered expression matches the target up to symbolic equivalence, and summarize black-box performance by test $R^2$ and expression size. We inject additive Gaussian noise into the *training* targets at a relative level of $\eta = 0.001$ (0.1%). Under this setting, GCN-SR exactly recovers 61 out of 100 equations (Table 2); additional noise-level and black-box results are provided in Appendix C.6.1.

### 4.6. Time Efficiency

We quantify the recovery–time tradeoff of GCN-SR by varying the *expression budget*, defined as the maximum total number of fitness evaluations allowed per run across the entire pipeline, while keeping all other settings fixed. For each budget, we evaluate GCN-SR on the Nguyen benchmark and report the recovery rate averaged over all tasks, together with the corresponding wall-clock time averaged over tasks. Fig. 4 shows that increasing the expression budget steadily improves recovery but exhibits diminishing returns at larger budgets, whereas runtime increases accordingly.

## 5. Limitations

GCN-SR also leaves room for further improvement. SPBT uses a fixed-depth scaffold to provide an explicit and batch-aligned tree representation. While this design enables efficient constraint-aware decoding, the scaffold size grows with the maximum depth, which may limit scalability for very large expressions. Future work may explore more

adaptive or sparse tree representations to further improve scalability while preserving explicit structural access.

## 6. Conclusion

GCN-SR performs symbolic regression via a graph-based tree generator trained in a strictly on-policy neural–evolutionary loop. SPBT provides a fixed-topology scaffold for batched decoding with explicit hierarchy and hard constraint enforcement. SWPG leverages GP only to construct similarity-weighted rewards, avoiding off-policy target matching. Experiments and ablations show consistent state-of-the-art recovery and validate both SPBT and SWPG. Future work will use large language models to extract domain-specific constraints from scientific literature.

## Impact Statement

This work contributes to symbolic regression by improving the efficient discovery of compact and interpretable mathematical expressions. It may benefit scientific modeling, engineering analysis, and data-driven tasks where explicit formulas are preferred over black-box predictors. However, automatically discovered expressions may capture spurious correlations and should not be interpreted as causal or physically valid without expert review, independent validation, and careful assessment of data quality.

## Acknowledgements

This work was supported by the National Science and Technology Major Project (2022ZD0117801), the National Natural Science Foundation of China (62306023), and the Beijing Nova Program (20240484490).

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

# A. Appendix: A. Pseudo code

---

**Algorithm 2** Sampling a Symbolic Expression via GCN on SPBT

---

**Require: Input:** Expression tree depth $D$; GCN layer count $L$; Symbol library $\mathcal{L} = \{\text{operators, variables, constants}\}$
**Ensure: Output:** A complete symbolic expression $\tau$

1:  **procedure** SAMPLEEXPRESSION($D, L, \mathcal{L}$)
2:      $|\mathcal{V}| \leftarrow 2^D - 1$ {Number of nodes in the perfect binary tree}
3:      Initialize feature matrix $X \in \mathbb{R}^{|\mathcal{V}| \times (|\mathcal{L}|+1)}$ with all rows as one-hot encoding of `<empty>`
4:      Construct adjacency matrix $A$ of the perfect binary tree
5:      Compute normalized adjacency matrix with self-loops:
        $\hat{A} = \tilde{D}^{-1/2}(A + I)\tilde{D}^{-1/2}$
6:      **for** $i = 0$ to $|\mathcal{V}| - 1$ **do** {Pre-order traversal of SPBT nodes}
7:          Compute node embeddings using GCN:
8:          $H^{(0)} \leftarrow XW^{(0)}$
9:          **for** $\ell = 0$ to $L - 1$ **do**
10:             $H^{(\ell+1)} \leftarrow \sigma(\hat{A}H^{(\ell)}W^{(\ell)})$
11:         **end for**
12:         Extract embedding for current node: $h_i \leftarrow H^L[i]$
13:         Compute logits: $z_i \leftarrow W_o^\top h_i$
14:         Apply softmax to get probability distribution: $\pi_i \leftarrow \text{softmax}(z_i)$
15:         Apply depth, syntactic, and semantic constraints to mask invalid tokens in $\pi_i$
16:         Sample token $a_i \sim \pi_i$
17:         Update feature matrix: $X[i] \leftarrow$ one-hot encoding of $a_i$
18:     **end for**
19:     Remove all `<empty>` tokens from the node assignments
20:     **return** resulting expression $\tau$
21: **end procedure**

---

---

**Algorithm 3** Similarity-Weighted Policy Gradient (SWPG)

---

**Require: Input:** Policy network parameters $\theta$ (GCN); Batch size $B$; Elite set size $K$; Genetic Programming (GP) configuration; Dataset $\mathcal{D} = \{(x_n, y_n)\}_{n=0}^{N-1}$

**Ensure: Output:** Updated policy parameters $\theta$

1:  **procedure** SWPGUPDATE($\theta, B, K, \text{GP}, \mathcal{D}$)
2:      **Phase 1: Neural Sampling**
3:      **for** $b = 0$ to $B - 1$ **do**
4:          Sample expression $\tau_\theta^b \sim \pi_\theta(\cdot)$ {Autoregressive sampling via GCN}
5:      **end for**
6:      **Phase 2: Genetic Programming Process**
7:      Set GP initial population $\leftarrow \{\tau_0, \ldots, \tau_{B-1}\}$
8:      Evolve population using GP with constrained mutation operations
9:      Combine neural samples and GP-evolved expressions into pool $\mathcal{P}$
10:     Select elite set $\mathcal{E} = \{\tau^{(1)}, \ldots, \tau^{(K)}\}$ {Top-K highest fitness from $\mathcal{P}$}
11:     **Phase 3: Similarity-Weighted Reward Calculation**
12:     **for** $b = 0$ to $B - 1$ **do**
13:         **for** $k = 1$ to $K$ **do**
14:             Compute structural similarity: $\lambda(\tau_\theta^b, \tau^{(k)}) = \frac{\text{LCS}(\tau_\theta^b, \tau^{(k)})}{|\tau_\theta^b|}$
15:             Evaluate fitness $R(\tau^{(k)})$ on dataset $\mathcal{D}$
16:         **end for**
17:         Compute similarity-weighted reward: $r_b = \frac{1}{K} \sum_{k=1}^{K} \lambda(\tau_\theta^b, \tau^{(k)}) \cdot R(\tau^{(k)})$
18:     **end for**
19:     Compute batch average reward: $\bar{r} = \frac{1}{B} \sum_{b=0}^{B-1} r_b$
20:     **Phase 4: Policy Update**
21:     $\nabla_\theta J_{\text{sim}}(\theta) \leftarrow 0$
22:     **for** $b = 0$ to $B - 1$ **do**
23:         Compute advantage: $A_b = r_b - \bar{r}$
24:         $\nabla_\theta J_{\text{sim}}(\theta) \leftarrow \nabla_\theta J_{\text{sim}}(\theta) + A_b \cdot \nabla_\theta \log \pi_\theta(\tau_\theta^b)$
25:     **end for**
26:     $\nabla_\theta J_{\text{sim}}(\theta) \leftarrow \frac{1}{B} \nabla_\theta J_{\text{sim}}(\theta)$
27:     $\theta \leftarrow \theta + \eta \cdot \nabla_\theta J_{\text{sim}}(\theta)$ {$\eta$: learning rate}
28:     **return** $\theta$
29: **end procedure**

---

# B. Appendix: B. Theoretical Proofs

We analyze the gradient estimator used in SWPG. Because GP is a discrete and non-differentiable procedure, the elite set $\mathcal{E}$ produced by GP is not analytically differentiable with respect to the policy parameters $\theta$. In implementation, SWPG executes GP first to obtain an elite set $\mathcal{E}$, and then performs a policy-gradient update while treating $\mathcal{E}$ as fixed within that update (i.e., stop-gradient on $\mathcal{E}$). Accordingly, SWPG optimizes a stop-gradient surrogate objective defined below, rather than claiming an unbiased estimator of the standard policy objective.

**Proposition B.1** (SWPG gradient estimator (stop-gradient surrogate)). *Fix an elite set $\mathcal{E}$ during a policy update and consider $J_{\mathrm{sim}}(\theta; \mathcal{E})$ below. Let $\{\tau_\theta^b\}_{b=1}^B \sim \pi_\theta$ be i.i.d. samples and $r_b = r(\tau_\theta^b, \mathcal{E})$. Then an on-policy Monte Carlo estimator of $\nabla_\theta J_{\mathrm{sim}}(\theta; \mathcal{E})$ is*

$$\nabla_\theta \hat{J}_{\mathrm{sim}}(\theta) = \frac{1}{B} \sum_{b=1}^B \left( r_b - \bar{r} \right) \nabla_\theta \log \pi_\theta(\tau_\theta^b), \tag{9}$$

*where $\bar{r} = \frac{1}{B} \sum_{b=1}^B r_b$ is a plug-in baseline used for variance reduction.*

*Proof.* **Step 1: Explicit term and implicit term.** Define the similarity-shaped objective that *includes* the dependence of GP output on $\theta$:

$$\widetilde{J}_{\mathrm{sim}}(\theta) := \mathbb{E}_{\tau \sim \pi_\theta}\Big[r(\tau, \mathcal{E}(\theta))\Big], \qquad r(\tau, \mathcal{E}) := \frac{1}{K} \sum_{k=1}^K \lambda(\tau, \tau^{(k)}) \, R(\tau^{(k)}), \tag{10}$$

where $\mathcal{E}(\theta)$ denotes the elite set returned by GP when initialized from policy samples under $\pi_\theta$.

By the chain rule, the total derivative decomposes into an *explicit term* and an *implicit term*:

$$\nabla_\theta \widetilde{J}_{\mathrm{sim}}(\theta) = \underbrace{\mathbb{E}_{\tau \sim \pi_\theta}[r(\tau, \mathcal{E}(\theta)) \, \nabla_\theta \log \pi_\theta(\tau)]}_{\text{explicit (score-function) term}} + \underbrace{\mathbb{E}_{\tau \sim \pi_\theta}\left[\left(\frac{\partial r(\tau, \mathcal{E})}{\partial \mathcal{E}}\Big|_{\mathcal{E}=\mathcal{E}(\theta)}\right)^\top \nabla_\theta \mathcal{E}(\theta)\right]}_{\text{implicit (through GP) term}}. \tag{11}$$

**Step 2: Stop-gradient surrogate used by SWPG.** The implicit term in (11) requires $\nabla_\theta \mathcal{E}(\theta)$, i.e., the derivative of the GP output with respect to $\theta$. This quantity is undefined in general because GP is a discrete, black-box combinatorial optimizer (selection, mutation, crossover), and does not admit an analytic gradient. Moreover, SWPG is implemented by first running GP to obtain $\mathcal{E}$ and then updating the policy while treating $\mathcal{E}$ as fixed; this corresponds to applying a stop-gradient operation on $\mathcal{E}$.

Therefore, SWPG optimizes the following *stop-gradient surrogate objective* with $\mathcal{E}$ fixed within the update:

$$J_{\mathrm{sim}}(\theta; \mathcal{E}) := \mathbb{E}_{\tau \sim \pi_\theta}\Big[r(\tau, \mathcal{E})\Big], \tag{12}$$

whose gradient contains only the explicit score-function component:

$$\nabla_\theta J_{\mathrm{sim}}(\theta; \mathcal{E}) = \mathbb{E}_{\tau \sim \pi_\theta}[r(\tau, \mathcal{E}) \, \nabla_\theta \log \pi_\theta(\tau)]. \tag{13}$$

Importantly, (13) is an *on-policy* gradient of the surrogate objective (12); GP affects the update only through the fixed set $\mathcal{E}$.

**Step 3: Monte Carlo estimator and plug-in baseline.** Let $\{\tau_\theta^b\}_{b=1}^B \sim \pi_\theta$ be i.i.d. on-policy samples, and define $r_b = r(\tau_\theta^b, \mathcal{E})$. A basic Monte Carlo estimator for (13) is:

$$\frac{1}{B} \sum_{b=1}^B r_b \, \nabla_\theta \log \pi_\theta(\tau_\theta^b). \tag{14}$$

SWPG subtracts the batch baseline $\bar{r} = \frac{1}{B} \sum_{b=1}^B r_b$ to reduce variance, yielding

$$\nabla_\theta \hat{J}_{\mathrm{sim}}(\theta) = \frac{1}{B} \sum_{b=1}^B (r_b - \bar{r}) \, \nabla_\theta \log \pi_\theta(\tau_\theta^b), \tag{15}$$

which is exactly the estimator stated in Proposition 3.1.

Note that because $\bar{r}$ depends on the same batch, this estimator is not an *exactly* unbiased estimator of (13); however it is a standard plug-in baseline. Indeed, using i.i.d. sampling and $\mathbb{E}_{\tau \sim \pi_\theta}[\nabla_\theta \log \pi_\theta(\tau)] = 0$, one can show

$$\mathbb{E}\left[\nabla_\theta \hat{J}_{\text{sim}}(\theta)\right] = \left(1 - \frac{1}{B}\right) \mathbb{E}_{\tau \sim \pi_\theta}[r(\tau, \mathcal{E}) \nabla_\theta \log \pi_\theta(\tau)], \tag{16}$$

so the bias is of order $O(\frac{1}{B})$ and vanishes as $B$ grows. This completes the justification of the estimator in Proposition 3.1. □

We next analyze the boundedness and variance of the Monte Carlo estimator $\nabla_\theta \hat{J}_{\text{sim}}(\theta)$ for the stop-gradient surrogate objective.

**Proposition B.2** (Boundedness and variance (stop-gradient surrogate)). *Assume: (i) $0 < R(\tau) \leq 1$ for any candidate $\tau$; (ii) $0 \leq \lambda(\tau, \tau^{(k)}) \leq 1$ for all $\tau$ and elites $\tau^{(k)} \in \mathcal{E}$; (iii) $\|\nabla_\theta \log \pi_\theta(\tau)\|_2 \leq C$ for some $C > 0$ and all $\tau$; (iv) $\mathcal{E}$ is fixed during the update. Then: (i) $\|\nabla_\theta \hat{J}_{sim}(\theta)\|_2 \leq C$; (ii) $\text{Var}\left(\nabla_\theta \hat{J}_{sim}(\theta)\right) \leq \frac{4C^2}{B}$.*

*Proof.* Let $g_b := \nabla_\theta \log \pi_\theta(\tau_\theta^b)$ and define

$$\hat{g} := \nabla_\theta \hat{J}_{\text{sim}}(\theta) = \frac{1}{B} \sum_{b=1}^{B} (r_b - \bar{r}) g_b, \qquad r_b = \frac{1}{K} \sum_{k=1}^{K} \lambda(\tau_\theta^b, \tau^{(k)}) R(\tau^{(k)}), \quad \bar{r} = \frac{1}{B} \sum_{b=1}^{B} r_b. \tag{17}$$

Throughout the proof, $\mathcal{E}$ is treated as fixed by assumption (iv), hence each $r_b$ is a function of $\tau_\theta^b$ only.

**(i) Boundedness.** From (i) and (ii), we have $0 \leq r_b \leq 1$, which implies $\bar{r} \in [0, 1]$ and therefore $|r_b - \bar{r}| \leq 1$. By (iii), $\|g_b\|_2 \leq C$. Hence,

$$\|\hat{g}\|_2 \leq \frac{1}{B} \sum_{b=1}^{B} |r_b - \bar{r}| \, \|g_b\|_2 \leq \frac{1}{B} \sum_{b=1}^{B} C = C. \tag{18}$$

**(ii) Variance control.** We use $\text{Var}(\hat{g}) := \text{Tr}(\text{Cov}(\hat{g}))$. Write $\hat{g}$ as the sum of two terms:

$$\hat{g} = \underbrace{\frac{1}{B} \sum_{b=1}^{B} r_b g_b}_{=:U} - \underbrace{\bar{r} \cdot \frac{1}{B} \sum_{b=1}^{B} g_b}_{=:V}. \tag{19}$$

By the inequality $\text{Var}(X - Y) \leq 2\,\text{Var}(X) + 2\,\text{Var}(Y)$ (for vector variance defined via trace of covariance), we have

$$\text{Var}(\hat{g}) \leq 2\,\text{Var}(U) + 2\,\text{Var}(V). \tag{20}$$

**Bound $\text{Var}(U)$.** Since $\{\tau_\theta^b\}_{b=1}^B$ are i.i.d. from $\pi_\theta$, the random vectors $\{r_b g_b\}_{b=1}^B$ are i.i.d. as well (with $\mathcal{E}$ fixed). Thus,

$$\text{Var}(U) = \text{Var}\left(\frac{1}{B} \sum_{b=1}^{B} r_b g_b\right) = \frac{1}{B} \text{Var}(r_1 g_1) \leq \frac{1}{B} \mathbb{E}\left[\|r_1 g_1\|_2^2\right]. \tag{21}$$

Using $0 \leq r_1 \leq 1$ and $\|g_1\|_2 \leq C$, we get $\|r_1 g_1\|_2^2 \leq C^2$, hence

$$\text{Var}(U) \leq \frac{C^2}{B}. \tag{22}$$

**Bound $\text{Var}(V)$.** First note that $\mathbb{E}_{\tau \sim \pi_\theta}[g(\tau)] = 0$, because

$$\mathbb{E}_{\tau \sim \pi_\theta}[\nabla_\theta \log \pi_\theta(\tau)] = \sum_\tau \pi_\theta(\tau) \nabla_\theta \log \pi_\theta(\tau) = \sum_\tau \nabla_\theta \pi_\theta(\tau) = \nabla_\theta \sum_\tau \pi_\theta(\tau) = \nabla_\theta 1 = 0. \tag{23}$$

Therefore, $\mathbb{E}\left[\frac{1}{B}\sum_{b=1}^{B} g_b\right] = 0$ and

$$\mathrm{Var}\left(\frac{1}{B}\sum_{b=1}^{B} g_b\right) = \frac{1}{B}\,\mathrm{Var}(g_1) \leq \frac{1}{B}\,\mathbb{E}\left[\|g_1\|_2^2\right] \leq \frac{C^2}{B}. \tag{24}$$

Since $\bar{r} \in [0,1]$, we have $\|V\|_2 \leq \left\|\frac{1}{B}\sum_{b=1}^{B} g_b\right\|_2$, hence

$$\mathrm{Var}(V) \leq \mathbb{E}\left[\|V\|_2^2\right] \leq \mathbb{E}\left[\left\|\frac{1}{B}\sum_{b=1}^{B} g_b\right\|_2^2\right] = \mathrm{Var}\left(\frac{1}{B}\sum_{b=1}^{B} g_b\right) \leq \frac{C^2}{B}. \tag{25}$$

Combining the two bounds,

$$\mathrm{Var}(\widehat{g}) \leq 2 \cdot \frac{C^2}{B} + 2 \cdot \frac{C^2}{B} = \frac{4C^2}{B}. \tag{26}$$

This proves Proposition 3.2. $\qquad\square$

# C. Appendix: C. Experiment Details

This appendix provides supplementary experimental details for GCN-SR. For readability, we structure Appendix C into six themed blocks: (C.1) local vs. global semantic constraints and their effect on symbolic recovery; (C.2) mechanism and implementation details; (C.3) experimental setup (reproducibility); (C.4) extended main results; (C.5) further analyses and ablations; and (C.6) additional experiments.

**Organization.** **C.1 Local vs. global semantic constraints** (Sec. C.1) defines the two constraint scopes and evaluates their impact on Nguyen under a matched evaluation budget. **C.2 Mechanism and implementation details** (Sec. C.2) describes the SPBT-SL representation and the semantic constraints enforced during batched decoding. **C.3 Experimental setup** (Sec. C.3) summarizes the tuned hyperparameters, hardware/software environment, the operator library, the symbolic recovery protocol, and the *baseline models*. **C.4 Extended main results** (Sec. C.4) details the benchmark problem set and reports extended per-task recovery tables. **C.5 Further analyses and ablations** (Sec. C.5) includes matched-capacity GCN vs. LSTM, SPBT depth and runtime, constant experiments, failure cases, and the recovery–budget trade-off on Nguyen. **C.6 Additional experiments** (Sec. C.6) reports SRBENCH results on both the Feynman suite and black-box tasks.

## C.1. Local vs. Global Semantic Constraints

This subsection clarifies two constraint scopes in symbolic expression generation and tests whether enforcing *global (hard)* constraints improves symbolic recovery under a matched evaluation budget.

**Local vs. global constraints.** We distinguish *local* constraints from *global (hard)* ones by the information required to validate a candidate during decoding. Local constraints only check *adjacent nesting* at the current expansion step (typically the parent–child relation), and thus can be implemented by a simple parent-conditioned decoding mask. In contrast, global constraints require *whole-path nesting* checks, i.e., whether a forbidden pattern occurs anywhere along the *entire ancestor chain* (and possibly across subtrees), which inherently depends on the evolving tree state. Sequence generators such as LSTM decode expressions as linear token streams; enforcing whole-path constraints would require maintaining an explicit tree structure for every partial sample during decoding, which is difficult to batch and undermines the standard vectorized sampling setup. Consequently, most sequence-based SR enforces only parent-based (local) masking (e.g., (Petersen et al., 2021b)). Therefore, in our comparison, LSTM is restricted to *local-only* constraints, while SPBT-based decoding can enforce both scopes, summarized in Table 4.

*Table 4.* Constraint implementability by representation and decoding interface.

| Method | Explicit tree state | Local constraints | Global constraints |
|---|:---:|:---:|:---:|
| GCN+SPBT (global) | ✓ | ✓ | ✓ |
| GCN+SPBT (local-only) | ✓ | ✓ | ✗ |
| LSTM (local-only) | ✗ | ✓ | ✗ |

**Experiment: effect of constraint scope.** We evaluate the above settings on the Nguyen benchmark using the same operator library and a matched evaluation budget. Each result is averaged over 10 independent runs and then aggregated across tasks. We report: (i) *recovery rate* (exact symbolic recovery), and (ii) *evaluation budget*, measured as the total number of candidate expressions evaluated until termination. To reflect the *actual cost of a successful recovery* and to avoid the statistics being dominated by runs that hit the maximum budget without recovery, the reported budget is computed *only over successful runs* (i.e., runs that exactly recover the target); failed runs are excluded from the budget averaging. Results are summarized in Table 5.

*Table 5.* Nguyen benchmark results under different constraint scopes.

| | GCN+SPBT (global) | | GCN+SPBT (local-only) | | LSTM (local-only) | |
|---|:---:|:---:|:---:|:---:|:---:|:---:|
| | Rec. (%) | Budget ($\times$ 1000) | Rec. (%) | Budget ($\times$ 1000) | Rec. (%) | Budget ($\times$ 1000) |
| **Average** | 57.50 | 423.11 | 48.33 | 597.22 | 51.67 | 903.74 |

In summary, enforcing global constraints consistently improves recovery rates and reduces the evaluation budget compared

to local-only constraints, particularly highlighting the importance of structural integrity during symbolic regression.

## C.2. Mechanism and Implementation Details

### C.2.1. SYMBOLIC PERFECT BINARY TREE WITH SELF-LOOPS (SPBT-SL)

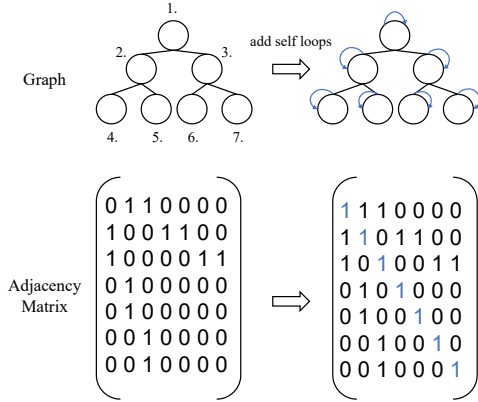

*Figure 5.* SPBT-SL and the corresponding adjacency matrix

### C.2.2. SEMANTIC CONSTRAINTS IN EXPRESSION GENERATION

To ensure that generated expressions are not only syntactically valid but also scientifically meaningful and numerically stable, GCN-SR enforces a set of domain-informed *semantic constraints* during autoregressive sampling. These constraints go beyond local grammar rules and target global structural patterns that are common sources of interpretability loss or numerical instability. In our implementation, the following semantic rules are strictly enforced in real time:

1. **No mutual nesting among trigonometric and inverse trigonometric functions**: Expressions such as $\sin(\cos(x))$, $\arcsin(\tan(x))$, or $\cos(x + \sin(x))$ are prohibited, as they are uncommon in benchmark ground-truth expressions and often lead to unnecessarily complex or overfitted forms.

2. **No invalid inverse-cancellation, overflow-prone exp–log nesting**: Forms like $\log(\exp(x))$, $\exp(\log(x))$, or deeply nested exponentials such as $\exp(\exp(x) + x)$ are disallowed due to algebraic redundancy or numerical overflow risk.

3. **No trigonometric subexpressions inside logarithms**: Expressions such as $\log(\sin(x) + \cos(x))$ are rejected because their arguments can become non-positive over the input domain, leading to numerical instability.

Importantly, these constraints are *global*: they depend on the hierarchical relationship between non-adjacent nodes in the expression tree (e.g., whether a $\cos$ appears anywhere in the subtree of a $\sin$). Sequence-based models such as LSTM or Transformer do not natively expose the full ancestor–descendant structure during autoregressive decoding; enforcing such global constraints therefore requires additional bookkeeping beyond the token sequence. In contrast, GCN-SR represents every candidate as a Symbolic Perfect Binary Tree (SPBT), enabling immediate traversal of ancestor–descendant relationships and real-time enforcement of subtree-based semantic rules—even in batched settings.

Beyond the three rules above, the framework naturally supports additional user-defined constraints. For instance, one can limit the nesting depth of multiplication/division chains, forbid division by expressions containing variables, or restrict the number of nested nonlinear operators. This flexibility allows GCN-SR to incorporate inductive biases from specific scientific domains, steering the search toward domain-consistent symbolic forms while maintaining end-to-end differentiability and training stability.

## C.3. Experimental Setup

### C.3.1. HYPERPARAMETERS

For our model, the following hyperparameter search space was considered: batch size $\in \{100, 200, 500, 1000, 1500, 2000\}$, learning rate $\in \{0.0001, 0.0003, 0.0005, 0.001\}$, number of GCN layers $\in \{1, 2, 3, 4\}$, and epsilon-greedy exploration rate $\in \{0.1, 0.2, 0.3, 0.4, 0.5\}$. Additionally, for the genetic programming component, the hyperparameter space included: population size $\in \{110, 210, 510, 1100, 1600, 2100\}$, tournament size $\in \{2, 3, 5, 10\}$, mutation probability $\in \{0.1, 0.2, 0.3, 0.4, 0.5\}$, crossover probability $\in \{0.2, 0.5, 0.8, 0.9\}$, and number of generations $\in \{10, 20, 30\}$. The full set of hyperparameters can be seen in Table 6. The depth of SPBT is set to 5, corresponding to a complete binary tree of depth 5, which can accommodate expressions with up to 31 tokens.

*Table 6.* Tuned hyperparameters

| Hyperparameter | Value |
| --- | --- |
| **GCN Parameters** | |
| SPBT-SL depth | 5 |
| Optimizer | Adam |
| GCN layers | 2 |
| GCN hidden state size | 32 |
| **Training Parameters** | |
| Maximum expressions | 3000000 |
| Batch size | 1000 |
| Learning rate | 0.0001 |
| epsilon-greedy | 0.5 |
| Exploration Bonus weight ($\beta$) | 0.001 |
| Early stop threshold | 1e-20 |
| Top-$K$ | 3 |
| **Genetic Programming** | |
| Population Size | 1100 |
| Crossover Probability | 0.8 |
| Mutation Probability | 0.2 |
| Number of Generations | 30 |
| Tournament Size | 5 |
| Maximum Tree Depth | 10 |

### C.3.2. HARDWARE/SOFTWARE ENVIRONMENT

The experiments were conducted on a platform equipped with an NVIDIA GeForce RTX 3080 GPU (10 GB VRAM), a 12th Gen Intel(R) Core(TM) i9-12900KF CPU (16 cores, 24 threads), 32 GB of system memory, and running Ubuntu 22.04 LTS.

### C.3.3. OPERATOR LIBRARY AND SYMBOLIC RECOVERY PROTOCOL

**Operator library and derived operators.** As stated in Table 7, all methods share the same allowable token set $\{+, -, \times, \div, \sin, \cos, \exp, \log, x, y\}$ (with $y$ removed for 1D tasks). For readability, some benchmark targets are written using a small number of *derived* operators (e.g., $\sqrt{\cdot}$, $\sinh$, $\cosh$, and fractional powers). During evaluation, we treat these symbols as deterministic shorthands and rewrite them into equivalent expressions over the allowable tokens, applying the same rewriting to both the candidate $\hat{f}$ and the ground truth $f^\star$ prior to symbolic equivalence checking. This canonicalization step is fixed and method-agnostic, ensuring it neither enlarges the search space nor advantages any particular method.

**Symbolic equivalence / recovery pipeline.** We compute symbolic recovery rate using a reproducible two-stage verification pipeline. Given a candidate expression $\hat{f}$ and the ground truth $f^\star$ (after applying the same rewriting when needed), we first perform a strict numerical screening on an independent validation set of $\mathcal{D}_{\text{val}}$ points (fixed seed) sampled from the task domain, and keep only candidates with $\mathrm{MSE}_{\text{val}}(\hat{f}, f^\star) \le \varepsilon$. This threshold serves as a *necessary* condition; for noise-free benchmarks, we set $\varepsilon$ extremely small (close to machine precision) since the data are generated directly from $f^\star$. For candidates passing the screening, we then use a computer algebra system (CAS) to check symbolic equivalence by simplifying $\hat{f} - f^\star$ and declare recovery only if the CAS confirms the identity. Therefore, a task is counted as recovered if and only if it passes *both* the strict numerical screening and the CAS-based equivalence check, yielding an objective and

consistent recovery metric across methods.

**Budget and wall-clock protocol.** We report the evaluation *budget* as the number of candidate expressions whose data-fitting loss is evaluated, i.e., one budget unit corresponds to evaluating one symbolic form on the training data. Following common practice in SR, the iterations of continuous constant fitting (when enabled) are *not* counted toward this budget. We also report *wall-clock time* as the single-run end-to-end runtime under a single-thread setting (fixing `OMP_NUM_THREADS=1` and disabling multi-threaded BLAS where applicable), excluding compilation and I/O overheads. Unless otherwise stated, wall-clock time includes all computations required to score a candidate, including constant fitting.

**Remark.** In the rare cases where CAS simplification is inconclusive, we rerun the equivalence check using additional random-point substitutions on the same domain (with a fixed seed). We additionally manually inspected *only* these inconclusive cases as a sanity check; this did not change any recovery outcomes.

### C.3.4. BASELINE MODELS

In this section, we provide additional details on the baseline configurations employed in both the main experiments and the SRBench evaluation.

**DSR**: Deep Symbolic Regression (DSR) (Petersen et al., 2021a) adopts a gradient-based reinforcement learning framework combined with a recurrent neural network (RNN) to model probability distributions over symbolic expressions. While it has been shown to perform well on longer and more complex expressions, its generalization behavior can vary across problem settings.

**NGGP**: Neural-Guided Genetic Programming (NGGP) (Mundhenk et al., 2021b) extends DSR by using the neural network's output as the initial population for genetic programming (GP). The GP component refines these candidate expressions, and the resulting optimized solutions are used to update the neural network, effectively leveraging GP as a source of high-quality expressions in an iterative manner.

**DGSR**: Deep Generative Symbolic Regression (DGSR) (Holt et al., 2023a) leverages pre-trained deep generative models to capture the intrinsic structural regularities and invariances (e.g., commutative properties) of mathematical expressions. The generative model provides a strong prior over the space of plausible equations, which is then refined through optimization techniques including genetic operations. This approach unifies several symbolic regression paradigms and has been reported to scale favorably with increasing numbers of input variables compared to reinforcement learning–based methods.

**RSRM**: The Reinforcement Symbolic Regression Machine (RSRM) (Xu & Sun, 2024a) is a reinforcement learning-based framework designed to recover complex mathematical equations from limited data. It integrates three key components: a Monte Carlo Tree Search (MCTS) agent for exploring expression trees, a Double Q-learning module to guide efficient exploitation through reward estimation, and a modulated sub-tree discovery mechanism that adaptively introduces new operators during the search.

**GPlearn**: gplearn (?) offers a flexible and efficient Python implementation of standard genetic programming for symbolic regression. It evolves expression trees using genetic operations—such as crossover, mutation, and selection—guided by fitness metrics like mean squared error.Despite its accessibility and ease of use, gplearn can exhibit variability across runs and may face scalability challenges on high-dimensional or structurally complex problems.

**GP-GOMEA**: GP-GOMEA (Virgolin, 2020) improves upon traditional genetic programming by incorporating the Gene-pool Optimal Mixing Evolutionary Algorithm (GOMEA). It utilizes linkage learning to identify dependencies among subtrees and applies structured, guided recombination to enhance search efficiency. This approach has been shown to achieve faster convergence and competitive solution quality on many symbolic regression benchmarks.

**AFP/AFP-FE**: Age-Fitness Pareto Optimization (AFP) (Schmidt & Lipson, 2010) is an evolutionary strategy that jointly optimizes for individual age and fitness to maintain population diversity and improve search dynamics. The AFP-FE variant incorporates Co-evolved Fitness Predictors (FE), which approximate fitness evaluations during evolution, potentially reducing computational overhead and accelerating convergence in many cases.

**uDSR**: uDSR (Landajuela et al., 2022) integrates components from DSR, AI Feynman, large-scale pre-training (LSPT), genetic programming (GP), and linear models (LM). It achieves strong performance in recovering formulas involving physical constants and shows a tendency toward polynomial-type expressions, albeit at the cost of increased computational

demand.

**AI Feynman**: AI Feynman (Udrescu & Tegmark, 2020a) combines neural network fitting with physics-inspired symbolic reasoning. By exploiting domain-specific properties such as symmetry, dimensional consistency, and functional decomposability, it recursively simplifies equations to recover compact and interpretable symbolic forms. This method has demonstrated strong effectiveness in certain scientific domains, particularly physics problems with specific structural properties.

**PySR**: PySR (Cranmer, 2023) is a high-performance symbolic regression library that employs evolutionary algorithms to discover accurate and parsimonious mathematical expressions. It leverages fast Julia-based computation while providing a Python interface, aiming to enable efficient and flexible symbolic modeling across diverse scientific applications. Its hybrid search strategy aims to balance prediction accuracy with expression simplicity.

We follow the recommended settings in the original papers as closely as possible, with minor adaptations to match the common token set and evaluation protocol.

### C.4. Extended Main Results

#### C.4.1. BENCHMARK PROBLEM SET

In this section, we give more details about the setting of benchmark tasks as shown in Table 7. We evaluate our method on three representative symbolic regression benchmarks with varying levels of difficulty and structural characteristics.

The **Nguyen** Benchmark is a widely adopted standard consisting of equations with one or two independent variables, sampled over predefined ranges. It primarily assesses the model's ability to recover simple to moderately complex functional forms.

The **Livermore** Benchmark introduces a higher level of difficulty, containing equations with complex structures such as high-degree polynomials, exponential terms, and trigonometric functions—making it well-suited for evaluating robustness and generalization.

The **R*** Benchmark focuses on rational expressions involving intricate combinations of numerators and denominators composed of multiple polynomials, posing challenges for accurate symbolic recovery. These benchmarks collectively provide a comprehensive assessment of symbolic regression performance across diverse functional domains.

#### C.4.2. EXTENDED PER-TASK RECOVERY TABLES

Tables 8, 9, and 10 report the recovery rates of individual problems across multiple benchmark suites for various algorithms, evaluated at data sizes of 20, 100, and 1000. Additional details regarding the performance of each model on specific expressions are provided therein.

### C.5. Further Analyses and Ablations

#### C.5.1. MATCHED-CAPACITY GCN VS. LSTM

Graph Convolutional Network–based Symbolic Regression (GCN-SR) and recurrent neural network–based approaches, such as those using LSTM, both frame symbolic regression as a policy optimization problem trained via policy gradients to autoregressively generate mathematical expressions. Both paradigms aim to efficiently explore the space of valid symbolic forms through neural-guided search. A key distinction lies in how syntactic structure is represented. LSTM-based models generate expressions as linear sequences using pre-order traversal—for example, encoding $\sin(x + \cos(x))$ as $[\sin, +, x, \cos, x]$—which offers simplicity and compatibility with standard sequence modeling tools. However, this representation does not preserve explicit hierarchical relationships among symbols, limiting the ability to enforce global semantic constraints during generation. In contrast, GCN-SR encodes expressions as Symbolic Perfect Binary Trees (SPBTs), a fixed-topology graph structure that retains full tree hierarchy and enables real-time evaluation of constraints that span non-local or deeply nested subexpressions.

To ensure a fair comparison, both models are configured to operate within comparable expression complexity budgets. Specifically, GCN-SR uses an SPBT of depth $D = 5$, supporting up to $2^5 - 1 = 31$ tokens, while the LSTM baseline adopts a maximum sequence length of 32 tokens—effectively aligning their generative capacity. Both methods are evaluated

on the Nguyen benchmark suite, with expression recovery assessed over multiple independent runs using exact symbolic equivalence. Training and sampling are performed with a batch size of 1,000 expressions per epoch on identical hardware. We report end-to-end wall-clock time per task (Table 11) and sampling throughput measured as wall-clock time per epoch (Table 12).

*Table 11.* Performance Comparison of GCN and LSTM Across Nguyen Tasks. Each task is evaluated with 10 independent runs. Recovery (%) is the success fraction over runs. Wall-clock / task (s) denotes the end-to-end time of one run (under the same evaluation budget). "–" indicates zero successful runs. Averages are computed across tasks (mean $\pm$ sample std).

| Nguyen | Expression | GCN | | LSTM | |
|---|---|---|---|---|---|
| | | Recovery (%) | Wall-clock / task (s) | Recovery (%) | Wall-clock / task (s) |
| 1 | $x^3 + x^2 + x$ | 100 | 92.33 | 100 | 81.91 |
| 2 | $x^4 + x^3 + x^2 + x$ | 20 | 511.29 | 100 | 151.23 |
| 3 | $x^5 + x^4 + x^3 + x^2 + x$ | 20 | 639.25 | 60 | 687.15 |
| 4 | $x^6 + x^5 + x^4 + x^3 + x^2 + x$ | 0 | – | 0 | – |
| 5 | $\sin(x^2)\cos(x) - 1$ | 30 | 38.93 | 0 | – |
| 6 | $\sin(x) + \sin(x + x^2)$ | 100 | 64.76 | 50 | 245.17 |
| 7 | $\log(x + 1) + \log(x^2 + 1)$ | 40 | 117.50 | 100 | 263.93 |
| 8 | $\sqrt{x}$ | 100 | 11.06 | 80 | 251.83 |
| 9 | $\sin(x) + \sin(y^2)$ | 100 | 58.32 | 50 | 199.80 |
| 10 | $2\sin(x)\cos(y)$ | 80 | 131.98 | 0 | – |
| 11 | $x^y$ | 100 | 4.88 | 80 | 18.00 |
| 12 | $x^4 - x^3 + \frac{1}{2}y^2 - y$ | 0 | – | 0 | – |
| **Average** | | **57.50$\pm$ 40.85** | **167.03$\pm$ 209.78** | **51.67$\pm$ 40.20** | **237.38$\pm$ 188.55** |

As shown in Table 11, GCN-SR achieves a higher average recovery rate (57.50% vs. 51.67%) and demonstrates consistent advantages on tasks involving nested or multi-variable structures. Notably, GCN-SR recovers Nguyen-6 ($\sin(x)+\sin(x+x^2)$) in 100% of runs compared to 50% for LSTM, and succeeds on Nguyen-10 ($2\sin(x)\cos(y)$) where LSTM yields no valid recovery. These improvements correlate with GCN-SR's ability to enforce global semantic rules during sampling—such as disallowing trigonometric operators from deeply nesting one another—which helps avoid uninterpretable or numerically unstable forms. Conversely, LSTM performs slightly better on simpler polynomial tasks like Nguyen-2 and Nguyen-7, where hierarchical structure plays a less critical role. In terms of computational cost, GCN-SR with $D = 5$ requires 0.3328 seconds per epoch versus 0.2350 seconds for LSTM (Table 12), reflecting additional overhead from graph message passing despite comparable output lengths. Together, these results suggest that explicit tree representation provides a measurable benefit for recovering structurally complex expressions, even when the target is relatively compact, at a small cost in sampling speed.

*Table 12.* Sampling throughput: average wall-clock time per epoch (batch size = 1,000 expressions). Uncertainties are standard deviations across repeated timing measurements.

| Method | GCN (D=4) | GCN (D=5) | GCN (D=6) | LSTM |
|---|---|---|---|---|
| Wall-clock / epoch (s) | $0.1197 \pm 0.0091$ | $0.3328 \pm 0.0173$ | $1.3521 \pm 0.0887$ | $0.2350 \pm 0.0058$ |

### C.5.2. IMPACT OF SPBT DEPTH AND RUNTIME CHARACTERISTICS

We further examine the impact of Symbolic Perfect Binary Tree (SPBT) depth on the overall performance of GCN-SR. As shown in Table 13, different depth settings exhibit a clear trade-off on the Nguyen benchmark. When the depth is too shallow (e.g., $D = 3$), the model lacks sufficient capacity to represent moderately complex expressions, resulting in 0% recovery on Nguyen-7 ($\log(x + 1) + \log(x^2 + 1)$), despite successful recovery on most simpler tasks. In contrast, both $D = 4$ and $D = 5$ achieve nearly perfect recovery across Tasks 1–11. (Note that Nguyen-12 is excluded from this analysis, as it achieved only 20% recovery at $D = 5$ with a runtime exceeding 1000 seconds.) Increasing depth further to $D = 6$ leads to a slight degradation in average recovery (97.3%), primarily due to a drop on Nguyen-5 ($\sin(x^2)\cos(x) - 1$), suggesting that excessive depth may introduce redundant structure that hinders learning. Focusing on the $D = 5$ configuration, the end-to-end runtime—encompassing GCN sampling, GP refinement, and SWPG updating—varies substantially across

tasks, ranging from **2.66 seconds** (Nguyen-1, $x^3 + x^2 + x$) to **177.35 seconds** (Nguyen-5, $\sin(x^2)\cos(x) - 1$). Most tasks complete within 10 to 60 seconds, indicating that the computational cost is closely tied to the structural complexity of the target expression.

*Table 13.* Symbolic regression performance on the Nguyen benchmark (Tasks 1–11). Recovery (%) is measured over 20 independent runs (each run uses a newly sampled dataset) under the same evaluation budget; Time (s) denotes end-to-end wall-clock time per run; "-" indicates zero successful runs. The Average row reports mean ± sample std across tasks. Note: Nguyen-12 is excluded because it only achieved 20% (D=5) and 10% (D=6) recovery rates with runtimes ≥ 1000 s, which would distort averages.

| Task | GCN-SR (D=3) | | GCN-SR (D=4) | | GCN-SR (D=5) | | GCN-SR (D=6) | |
|---|---|---|---|---|---|---|---|---|
| | Recovery (%) | Time (s) | Recovery (%) | Time (s) | Recovery (%) | Time (s) | Recovery (%) | Time (s) |
| Nguyen-1 | 100 | 3.15 | 100 | 1.29 | 100 | 2.66 | 100 | 18.14 |
| Nguyen-2 | 100 | 4.93 | 100 | 11.33 | 100 | 12.48 | 100 | 78.96 |
| Nguyen-3 | 100 | 35.53 | 100 | 25.32 | 100 | 15.14 | 100 | 116.89 |
| Nguyen-4 | 100 | 257.79 | 100 | 47.17 | 100 | 148.40 | 100 | 312.50 |
| Nguyen-5 | 100 | 51.09 | 100 | 132.36 | 100 | 177.35 | 70 | 442.50 |
| Nguyen-6 | 100 | 3.70 | 100 | 2.92 | 100 | 7.95 | 100 | 77.47 |
| Nguyen-7 | 0 | - | 100 | 87.06 | 100 | 56.75 | 100 | 71.53 |
| Nguyen-8 | 100 | 643.00 | 100 | 10.42 | 100 | 22.66 | 100 | 32.23 |
| Nguyen-9 | 100 | 1.76 | 100 | 3.40 | 100 | 4.23 | 100 | 19.75 |
| Nguyen-10 | 100 | 3.94 | 100 | 2.77 | 100 | 6.72 | 100 | 24.43 |
| Nguyen-11 | 100 | 101.93 | 100 | 5.72 | 100 | 6.61 | 100 | 13.96 |
| **Average** | **90.9±30.2** | **101.0±195.5** | **100.0±0.0** | **29.98±42.87** | **100.0±0.0** | **41.90±62.01** | **97.3±9.1** | **109.85±139.39** |

### C.5.3. CONSTANT EXPERIMENTS (JIN)

Table 14 compares the RMSE performance of DSR, NGGP, RSRM, and GCN-SR on the Jin benchmark set. NGGP, RSRM, and GCN-SR yield RMSE values close to zero (smaller than $10^{-8}$) on most tasks, indicating superior accuracy compared to DSR. For all methods, continuous constants were refined using BFGS following the same protocol as DSR (Petersen et al., 2021a).

*Table 14.* Comparison of mean root-mean-square error (RMSE) of different methods

| Name | Expression | DSR | NGGP | RSRM | GCN-SR |
|---|---|---|---|---|---|
| Jin-1 | $2.5x^4 - 1.3x^3 + 0.5y^2 - 1.7y$ | 0.46 | 0 | 0 | 0 |
| Jin-2 | $8.0x^2 + 8.0y^3 - 15.0$ | 0 | 0 | 0 | 0 |
| Jin-3 | $0.2x^3 + 0.5y^3 - 1.2y - 0.5x$ | 0.00052 | 0 | 0 | 0 |
| Jin-4 | $1.5\exp(x) + 5.0\cos(y)$ | 0.00014 | 0 | 0 | 0 |
| Jin-5 | $6.0\sin(x)\cos(y)$ | 0 | 0 | 0 | 0 |
| Jin-6 | $1.35x^2 + 5.5\sin(x - 1.0)^2$ | 2.23 | 0 | 0 | 0 |
| Average | | 0.45 | 0 | 0 | 0 |

### C.5.4. FAILURE CASES AND LOCAL OPTIMA

Table 15 shows representative failure cases where GCN-SR recovers expressions that are numerically close to the target but symbolically incorrect. Notably, for Livermore-7 and Livermore-8, the recovered expressions achieve extremely low MSE ($< 10^{-7}$) yet differ in structure from the ground truth, indicating that the search converged to a local optimum that fits the data well but fails to capture the true symbolic form.

*Table 15.* Analysis of Failed Symbolic Regression Attempts

| Task | Target Expression | Recovered Expression | MSE | Nodes | Depth |
|---|---|---|---|---|---|
| Nguyen-12 | $x_1^4 - x_1^3 + \frac{1}{2}x_2^2 - x_2$ | $x_2 \cdot (\sin(x_2) - x_1 - \cos(x_1))$ | $1.8006 \times 10^{-2}$ | 9 | 5 |
| Livermore-7 | $\sinh(x_1)$ | $\exp(x_1) - x_1^2 - \cos(x_1)$ | $3.8004 \times 10^{-7}$ | 9 | 4 |
| Livermore-8 | $\cosh(x_1)$ | $x_1^2 + \cos(x_1)$ | $2.1776 \times 10^{-7}$ | 6 | 3 |

We record recovery-rate curves of GCN-SR as a function of the number of sampled expressions on Nguyen over 20 runs. Most tasks are recovered within roughly $10^6$ sampled expressions, while harder targets require larger budgets.

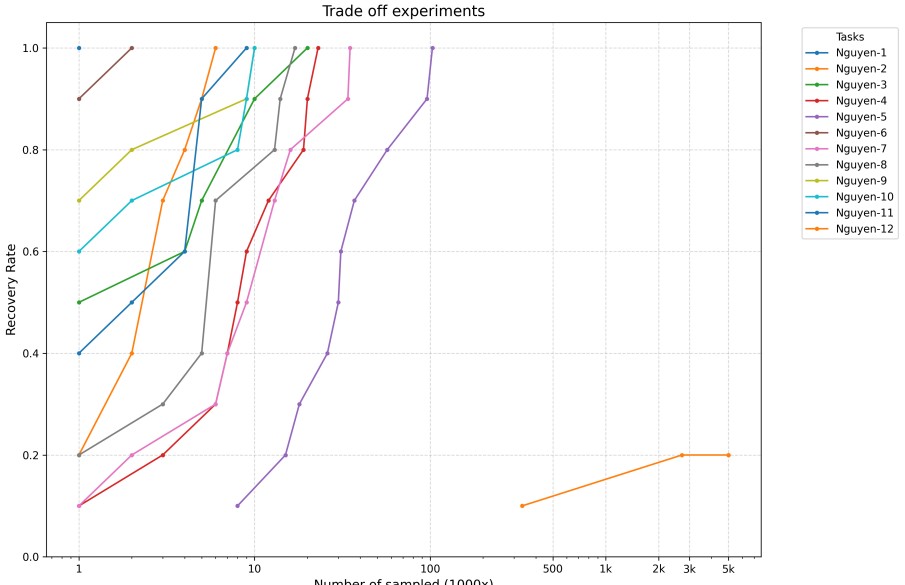

*Figure 6.* Trade-off between recovery rate and number of sampled expressions in Nguyen Benchmark

## C.6. Additional Experiments

### C.6.1. SRBENCH RESULTS

**Feynman suite (white-box).** The SRBENCH Feynman Equations suite comprises 100 closed-form physics equations paired with synthetic measurements (Feynman et al., 2015; La Cava et al., 2021). Because the ground-truth expressions are available, we evaluate methods using the benchmark *solution rate*, i.e., the fraction of equations for which the recovered expression is symbolically equivalent to the target (Sec. 4.5). In this white-box setting, purely predictive metrics such as $R^2$ can be insufficient: a method may achieve high $R^2$ by fitting numerically flexible yet scientifically uninformative forms (e.g., over-parameterized expansions with optimized constants) without recovering the underlying law. Therefore, exact symbolic recovery is the appropriate criterion for assessing scientific discovery on the Feynman suite. Following the official SRBENCH protocol, we inject additive Gaussian noise into the *training* targets and report solution rates under four relative noise levels $\eta \in \{0, 0.001, 0.01, 0.1\}$. Table 2 reports results at $\eta = 0.001$ (main text), while Fig. 7(a) summarizes performance across all noise levels. We use the same evaluation budget setting for all compared methods.

**Black-box tasks.** SRBENCH also includes black-box regression tasks where the underlying functional forms are unknown (La Cava et al., 2021). Since exact recovery is not defined, we evaluate the *accuracy–complexity* tradeoff of the discovered expressions. Following the benchmark protocol, we report test-set $R^2$ as predictive performance and measure complexity by expression size (number of nodes). Figure 7(b–c) summarizes the black-box performance in terms of complexity and test $R^2$, and Fig. 8 further visualizes the corresponding Pareto frontiers. Overall, GCN-SR achieves competitive frontier points, indicating strong predictive accuracy while maintaining moderate expression complexity.

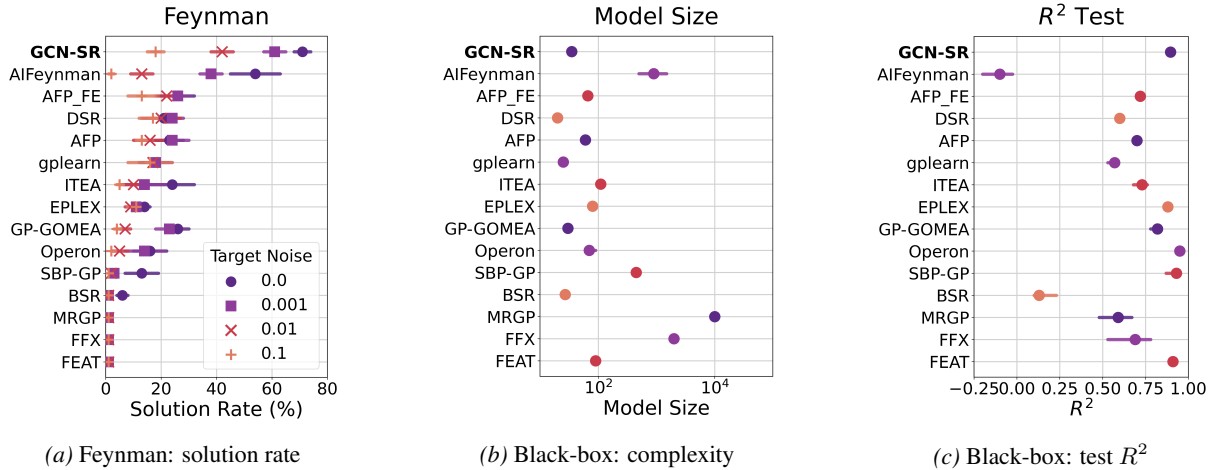

*(a)* Feynman: solution rate      *(b)* Black-box: complexity      *(c)* Black-box: test $R^2$

*Figure 7.* SRBENCH results. (a) Feynman suite (white-box): solution rate under four training-target noise levels $\eta \in \{0, 0.001, 0.01, 0.1\}$. (b–c) Black-box tasks: (b) expression complexity (number of nodes) and (c) predictive performance (test $R^2$) across methods.

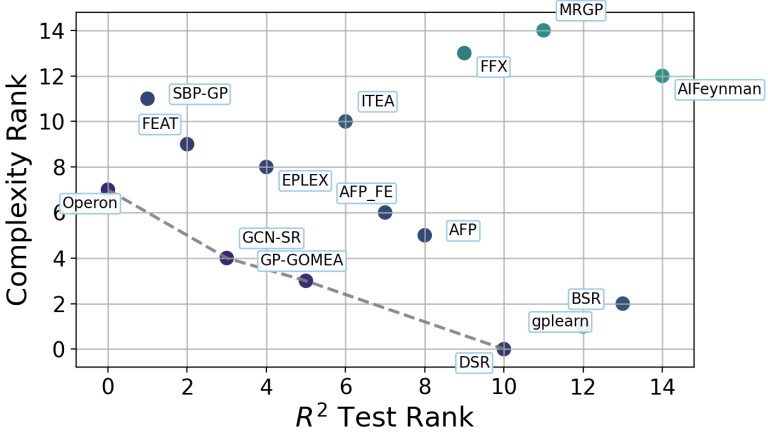

*Figure 8.* Pareto frontiers on black-box SRBENCH, illustrating the accuracy–complexity tradeoff across methods.

*Table 7.* Benchmark symbolic regression problem specifications. Input variables are denoted by $x$ and/or $y$, training and test datasets use different random seeds. $U(a, b)$ denotes sampled points evenly spaced between $a$ and $b$ for each input variable; training and test datasets use the same points. All benchmark problems use the following set of allowable tokens: $\{+, -, \times, \div, \sin, \cos, \exp, \log, x, y\}$ (y is excluded for single-dimensional datasets). (Several benchmark formulas are written using derived operators for readability; these are deterministically rewritten into equivalent expressions over the allowable tokens during evaluation; see Appendix C.3.3.)

| Name | Expression | Dataset |
|------|-----------|---------|
| Nguyen-1 | $x^3 + x^2 + x$ | $U(-1, 1)$ |
| Nguyen-2 | $x^4 + x^3 + x^2 + x$ | $U(-1, 1)$ |
| Nguyen-3 | $x^5 + x^4 + x^3 + x^2 + x$ | $U(-1, 1)$ |
| Nguyen-4 | $x^6 + x^5 + x^4 + x^3 + x^2 + x$ | $U(-1, 1)$ |
| Nguyen-5 | $\sin(x^2)\cos(x) - 1$ | $U(-1, 1)$ |
| Nguyen-6 | $\sin(x) + \sin(x + x^2)$ | $U(-1, 1)$ |
| Nguyen-7 | $\log(x + 1) + \log(x^2 + 1)$ | $U(0, 2)$ |
| Nguyen-8 | $\sqrt{x}$ | $U(0, 4)$ |
| Nguyen-9 | $\sin(x) + \sin(y^2)$ | $U(0, 1)$ |
| Nguyen-10 | $2\sin(x)\cos(y)$ | $U(0, 1)$ |
| Nguyen-11 | $x^y$ | $U(0, 1)$ |
| Nguyen-12 | $x^4 - x^3 + \frac{1}{2}y^2 - y$ | $U(0, 1)$ |
| R*-1 | $\frac{(x+1)^3}{x^2 - x + 1}$ | $U(-10, 10)$ |
| R*-2 | $\frac{x^5 - 3x^3 + 1}{x^2 + 1}$ | $U(-10, 10)$ |
| R*-3 | $\frac{x^6 + x^5}{x^4 + x^3 + x^2 + x + 1}$ | $U(-10, 10)$ |
| Livermore-1 | $\frac{1}{3} + x + \sin(x^2)$ | $U(-1, 1)$ |
| Livermore-2 | $\sin(x^2)\cos(x) - 2$ | $U(-1, 1)$ |
| Livermore-3 | $\sin(x^3)\cos(x^2) - 1$ | $U(-1, 1)$ |
| Livermore-4 | $\log(x + 1) + \log(x^2 + 1) + \log(x)$ | $U(0, 2)$ |
| Livermore-5 | $x^4 - x^3 + x^2 - y$ | $U(0, 1)$ |
| Livermore-6 | $4x^4 + 3x^2 + 2x + x$ | $U(-1, 1)$ |
| Livermore-7 | $\sinh(x)$ | $U(-1, 1)$ |
| Livermore-8 | $\cosh(x)$ | $U(-1, 1)$ |
| Livermore-9 | $x^9 + x^8 + x^7 + x^6 + x^5 + x^4 + x^3 + x^2 + x$ | $U(-1, 1)$ |
| Livermore-10 | $6\sin(x)\cos(y)$ | $U(0, 1)$ |
| Livermore-11 | $\frac{x^4}{y + x}$ | $U(0, 1)$ |
| Livermore-12 | $\frac{x^5}{y^3}$ | $U(-1, 1)$ |
| Livermore-13 | $x^{\frac{1}{3}}$ | $U(0, 1)$ |
| Livermore-14 | $x^3 + x^2 + x + \sin(x) + \sin(x^2)$ | $U(-1, 1)$ |
| Livermore-15 | $x^{1/5}$ | $U(0, 4)$ |
| Livermore-16 | $x^{2/5}$ | $U(0, 4)$ |
| Livermore-17 | $4\sin(x)\cos(y)$ | $U(0, 1)$ |
| Livermore-18 | $\sin(x^2)\cos(x) - 5$ | $U(-1, 1)$ |
| Livermore-19 | $x^5 + x^4 + x^2 + x$ | $U(-1, 1)$ |
| Livermore-20 | $\exp(-x^2)$ | $U(-1, 1)$ |
| Livermore-21 | $x^8 + x^7 + x^6 + x^5 + x^4 + x^3 + x^2 + x$ | $U(-1, 1)$ |
| Livermore-22 | $\exp(-0.5x^2)$ | $U(-1, 1)$ |

*Table 8.* Comparison of different algorithms on Nguyen, Livermore, and R datasets with $N{=}20$ data points. Each entry reports recovery rate (%) over $J{=}20$ independent runs, where each run uses a newly sampled dataset. All methods are evaluated under the same evaluation budget (total number of evaluated candidate expressions per run). Averages are computed across tasks within each benchmark.

| | GP | DSR | DGSR | NGGP | RSRM | GCN-SR |
|---|---|---|---|---|---|---|
| Nguyen-1 | 100% | 100% | 100% | 100% | 100% | 100% |
| Nguyen-2 | 100% | 100% | 100% | 100% | 100% | 100% |
| Nguyen-3 | 100% | 100% | 100% | 100% | 100% | 100% |
| Nguyen-4 | 100% | 100% | 100% | 100% | 100% | 100% |
| Nguyen-5 | 50% | 70% | 100% | 100% | 100% | 100% |
| Nguyen-6 | 90% | 100% | 100% | 100% | 100% | 100% |
| Nguyen-7 | 0% | 40% | 0% | 100% | 0% | 100% |
| Nguyen-8 | 0% | 100% | 100% | 100% | 100% | 100% |
| Nguyen-9 | 100% | 100% | 100% | 100% | 100% | 100% |
| Nguyen-10 | 90% | 100% | 100% | 100% | 100% | 100% |
| Nguyen-11 | 10% | 100% | 50% | 100% | 100% | 100% |
| Nguyen-12 | 0% | 0% | 0% | 0% | 0% | 20% |
| **Nguyen Average** | 61.6% | 84.2% | 79.2% | 91.7% | 83.3% | 93.3% |
| Livermore-1 | 10% | 0% | 100% | 100% | 100% | 100% |
| Livermore-2 | 0% | 90% | 100% | 100% | 100% | 100% |
| Livermore-3 | 10% | 40% | 100% | 100% | 100% | 100% |
| Livermore-4 | 50% | 80% | 100% | 100% | 100% | 100% |
| Livermore-5 | 50% | 0% | 90% | 50% | 90% | 100% |
| Livermore-6 | 20% | 40% | 100% | 100% | 70% | 100% |
| Livermore-7 | 0% | 0% | 0% | 0% | 0% | 0% |
| Livermore-8 | 0% | 0% | 0% | 0% | 0% | 20% |
| Livermore-9 | 0% | 0% | 50% | 20% | 0% | 80% |
| Livermore-10 | 0% | 0% | 30% | 0% | 90% | 100% |
| Livermore-11 | 80% | 90% | 100% | 100% | 100% | 100% |
| Livermore-12 | 0% | 20% | 100% | 100% | 90% | 100% |
| Livermore-13 | 10% | 70% | 100% | 100% | 100% | 100% |
| Livermore-14 | 60% | 20% | 100% | 100% | 100% | 100% |
| Livermore-15 | 0% | 0% | 100% | 100% | 100% | 100% |
| Livermore-16 | 0% | 20% | 60% | 70% | 80% | 100% |
| Livermore-17 | 0% | 0% | 60% | 90% | 90% | 100% |
| Livermore-18 | 0% | 0% | 100% | 80% | 90% | 80% |
| Livermore-19 | 60% | 100% | 100% | 100% | 100% | 100% |
| Livermore-20 | 30% | 100% | 100% | 100% | 100% | 100% |
| Livermore-21 | 0% | 0% | 100% | 100% | 60% | 100% |
| Livermore-22 | 0% | 0% | 10% | 100% | 70% | 100% |
| **Livermore Average** | 17.2% | 30.5% | 77.2% | 77.7% | 78.6% | 90.0% |
| R*-1 | 0% | 0% | 0% | 0% | 100% | 100% |
| R*-2 | 0% | 0% | 0% | 0% | 100% | 100% |
| R*-3 | 0% | 0% | 30% | 40% | 90% | 100% |
| **R Average** | 0% | 0% | 10.0% | 13.3% | 97.0% | 100% |

*Table 9.* Comparison of different algorithms on Nguyen, Livermore, and R datasets with $N{=}100$ data points. Each entry reports recovery rate (%) over $J{=}20$ independent runs, where each run uses a newly sampled dataset. All methods are evaluated under the same evaluation budget (total number of evaluated candidate expressions per run). Averages are computed across tasks within each benchmark.

| | GP | DSR | DGSR | NGGP | RSRM | GCN-SR |
|---|---|---|---|---|---|---|
| Nguyen-1 | 100% | 100% | 100% | 100% | 100% | 100% |
| Nguyen-2 | 100% | 100% | 100% | 100% | 100% | 100% |
| Nguyen-3 | 100% | 100% | 100% | 100% | 100% | 100% |
| Nguyen-4 | 100% | 100% | 100% | 100% | 100% | 100% |
| Nguyen-5 | 60% | 90% | 100% | 100% | 100% | 100% |
| Nguyen-6 | 80% | 100% | 100% | 100% | 100% | 100% |
| Nguyen-7 | 0% | 50% | 10% | 100% | 0% | 100% |
| Nguyen-8 | 10% | 90% | 100% | 100% | 100% | 100% |
| Nguyen-9 | 100% | 100% | 100% | 100% | 100% | 100% |
| Nguyen-10 | 90% | 100% | 100% | 100% | 100% | 100% |
| Nguyen-11 | 10% | 100% | 70% | 100% | 100% | 100% |
| Nguyen-12 | 0% | 0% | 0% | 0% | 20% | 20% |
| **Nguyen Average** | 62.5% | 85.8% | 81.7% | 91.7% | 85.0% | 93.3% |
| Livermore-1 | 40% | 0% | 100% | 100% | 100% | 100% |
| Livermore-2 | 0% | 100% | 100% | 100% | 100% | 100% |
| Livermore-3 | 60% | 60% | 100% | 100% | 100% | 100% |
| Livermore-4 | 50% | 100% | 100% | 100% | 100% | 100% |
| Livermore-5 | 60% | 0% | 90% | 50% | 100% | 100% |
| Livermore-6 | 10% | 0% | 100% | 100% | 70% | 100% |
| Livermore-7 | 0% | 0% | 0% | 0% | 0% | 10% |
| Livermore-8 | 0% | 0% | 0% | 0% | 0% | 10% |
| Livermore-9 | 0% | 0% | 60% | 40% | 0% | 80% |
| Livermore-10 | 0% | 0% | 50% | 10% | 100% | 100% |
| Livermore-11 | 80% | 100% | 100% | 100% | 100% | 100% |
| Livermore-12 | 0% | 0% | 90% | 80% | 100% | 100% |
| Livermore-13 | 30% | 30% | 100% | 100% | 100% | 100% |
| Livermore-14 | 40% | 20% | 100% | 100% | 100% | 100% |
| Livermore-15 | 0% | 0% | 100% | 100% | 100% | 100% |
| Livermore-16 | 0% | 0% | 30% | 100% | 100% | 100% |
| Livermore-17 | 0% | 0% | 80% | 60% | 100% | 100% |
| Livermore-18 | 0% | 0% | 100% | 70% | 100% | 90% |
| Livermore-19 | 60% | 100% | 100% | 100% | 100% | 100% |
| Livermore-20 | 30% | 100% | 100% | 100% | 100% | 100% |
| Livermore-21 | 50% | 30% | 80% | 80% | 100% | 100% |
| Livermore-22 | 0% | 0% | 50% | 90% | 60% | 100% |
| **Livermore Average** | 23.2% | 29.0% | 78.6% | 76.3% | 83.2% | 90.5% |
| R*-1 | 0% | 0% | 50% | 0% | 100% | 100% |
| R*-2 | 0% | 0% | 0% | 0% | 100% | 100% |
| R*-3 | 0% | 0% | 70% | 100% | 100% | 100% |
| **R Average** | 0% | 0% | 40.0% | 33.3% | 100% | 100% |

*Table 10.* Comparison of different algorithms on Nguyen, Livermore, and R datasets with $N{=}1000$ data points. Each entry reports recovery rate (%) over $J{=}20$ independent runs, where each run uses a newly sampled dataset. All methods are evaluated under the same evaluation budget (total number of evaluated candidate expressions per run). Averages are computed across tasks within each benchmark.

| | GP | DSR | DGSR | NGGP | RSRM | GCN-SR |
|---|---|---|---|---|---|---|
| Nguyen-1 | 100% | 100% | 100% | 100% | 100% | 100% |
| Nguyen-2 | 100% | 100% | 100% | 100% | 100% | 100% |
| Nguyen-3 | 100% | 100% | 100% | 100% | 100% | 100% |
| Nguyen-4 | 100% | 100% | 100% | 100% | 100% | 100% |
| Nguyen-5 | 50% | 100% | 100% | 100% | 100% | 100% |
| Nguyen-6 | 90% | 100% | 100% | 100% | 100% | 100% |
| Nguyen-7 | 0% | 50% | 10% | 100% | 90% | 100% |
| Nguyen-8 | 10% | 100% | 100% | 100% | 100% | 100% |
| Nguyen-9 | 100% | 100% | 100% | 100% | 100% | 100% |
| Nguyen-10 | 100% | 100% | 100% | 100% | 100% | 100% |
| Nguyen-11 | 0% | 100% | 80% | 100% | 100% | 100% |
| Nguyen-12 | 0% | 0% | 0% | 0% | 20% | 20% |
| **Nguyen Average** | 62.5% | 87.5% | 82.5% | 91.7% | 92.5% | 93.3% |
| Livermore-1 | 10% | 0% | 90% | 100% | 100% | 100% |
| Livermore-2 | 0% | 100% | 100% | 100% | 100% | 100% |
| Livermore-3 | 60% | 50% | 100% | 100% | 100% | 100% |
| Livermore-4 | 50% | 100% | 100% | 100% | 100% | 100% |
| Livermore-5 | 10% | 0% | 100% | 90% | 100% | 100% |
| Livermore-6 | 30% | 50% | 100% | 100% | 70% | 100% |
| Livermore-7 | 0% | 0% | 0% | 0% | 0% | 0% |
| Livermore-8 | 0% | 0% | 0% | 0% | 0% | 20% |
| Livermore-9 | 0% | 0% | 60% | 40% | 40% | 100% |
| Livermore-10 | 0% | 0% | 70% | 10% | 100% | 100% |
| Livermore-11 | 80% | 90% | 100% | 100% | 100% | 100% |
| Livermore-12 | 0% | 0% | 100% | 50% | 100% | 100% |
| Livermore-13 | 10% | 30% | 100% | 100% | 100% | 100% |
| Livermore-14 | 60% | 20% | 100% | 100% | 100% | 100% |
| Livermore-15 | 0% | 0% | 90% | 100% | 100% | 100% |
| Livermore-16 | 0% | 0% | 10% | 90% | 100% | 100% |
| Livermore-17 | 0% | 0% | 70% | 100% | 100% | 100% |
| Livermore-18 | 0% | 0% | 90% | 80% | 80% | 100% |
| Livermore-19 | 60% | 100% | 100% | 100% | 100% | 100% |
| Livermore-20 | 30% | 100% | 100% | 100% | 100% | 100% |
| Livermore-21 | 30% | 20% | 100% | 100% | 100% | 100% |
| Livermore-22 | 0% | 0% | 40% | 90% | 90% | 100% |
| **Livermore Average** | 19.5% | 30.0% | 78.2% | 78.6% | 85.5% | 91.% |
| R*-1 | 0% | 40% | 60% | 0% | 100% | 100% |
| R*-2 | 0% | 0% | 0% | 0% | 100% | 100% |
| R*-3 | 0% | 0% | 80% | 100% | 100% | 100% |
| **R Average** | 0% | 13.3% | 46.6% | 33.3% | 100% | 100% |

