# OpenReview forum: "Neural–Evolutionary Symbolic Regression with Global Constraints: Constraint-Aware Decoding and Reward Shaping"
_ICML.cc/2026/Conference — ICML 2026 regular_

### Official Review · Reviewer_uRiL · 2026-03-02

**Soundness:** 3
**Presentation:** 3
**Significance:** 2
**Originality:** 2
**Overall Recommendation:** 4
**Confidence:** 3

**Summary:**

This paper improves symbolic regression by using a graph-based tree scaffold for better constraint enforcement and a similarity-weighted reward mechanism to stabilize the integration of neural generation with evolutionary search.

**Compliance With Llm Reviewing Policy:**

Affirmed.

**Final Justification:**

I think this paper provided solid technical contributions. With more baseline method implemented for comparison, the paper will be stronger for publication.

**Key Questions For Authors:**

1. Refer to point #1, I am happy to increase the score if the existing benchmarks are carefully discussed.

2. Overall, I think this paper provided a simple but effective way for SR: SPBT, which the results show a quite significant improvement compared with the state-of-art. However, with method simpler, it is important to clarify how the method can extend to a more complicate settings: complex formula, large tree depth, etc. I would like to further increase the score if this point is clarified.

**Limitations:**

Limitation is not discussed. It is suggestive to add one to two points.

**Strengths And Weaknesses:**

Strength

(1) Unlike standard neural approaches that linearize expression trees into token sequences, GCN-SR uses a Symbolic Perfect Binary Tree (SPBT) scaffold. This preserves the full tree hierarchy during generation.

(2) The framework introduces Similarity-Weighted Policy Gradient (SWPG). This method uses Genetic Programming (GP) to construct reward signals which can mitigates the "distribution mismatch".

Weakness

(1) The result section is the main concerns of this paper. To verify the effectiveness of SPBT, I think the comparison with other methods need to be more careful. However, a lot of popular benchmark is missing here for example:
 a. NeurSR: https://proceedings.mlr.press/v139/biggio21a/biggio21a.pdf
 b. FEAT-KD: https://openreview.net/forum?id=ApVH0G4l6P
 c. PySR: Cranmer, M. (2023). Interpretable machine learning for science with PySR and SymbolicRegression. jl. arXiv preprint arXiv:2305.01582.
 d. Operon: Burlacu, B., Kronberger, G., & Kommenda, M. (2020, July). Operon C++ an efficient genetic programming framework for symbolic regression. In Proceedings of the 2020 genetic and evolutionary computation conference companion (pp. 1562-1570).

(2) This method replies on maximum tree depth, which may limit the space of discovery. So how to set this number may need more in-depth discussion.

(3) The sampling cost of this method is one concern. The computation cost of each decoding step may increase exponentially with the tree size, due to a lot ot "empty" node output.

(4) Based on the previous two points, I would also like to point out similar work that moves away from simple sequence generation to capture complex tree structure [1]. However, it seems that the works that can freely search the tree structure is not discussed much in this paper. And it would be suggestive to provide more clarification of the added values using SPBT.

[1] https://openreview.net/forum?id=r3fzEWnaY4&noteId=pcw9MmQIs3

---

> ### Author Rebuttal · Authors · 2026-03-31
>
> We thank the reviewer for the helpful comments and for pointing out this related line of work. Below we address the specific points.
>
> Q1.
> We agree that benchmark coverage and comparability should be explained more clearly. NeurSR belongs to the large-scale pre-trained generative line of neural symbolic regression. In the main paper, the directly comparable trained neural baseline is DGSR, which is more recent and belongs to the same broad family of generative neural SR methods; thus NeurSR was not intentionally omitted, but separated from the most directly comparable baseline family. FEAT-KD is closer to symbolic feature distillation / interpretable regression, where the final prediction is a weighted combination of symbolic features rather than direct recovery of a single target expression in the standard white-box SR setting.
>
> To address this concern more directly, we additionally ran N=1000 experiments for NeurSR and FEAT-KD on the same white-box benchmarks. The results are: Nguyen 76.2±36.9 / 91.7±23.0, Livermore 35.7±27.1 / 76.46±14.1, and R* 41.7±37.2 / 85.5±35.7 for NeurSR / FEAT-KD, respectively. These results suggest that FEAT-KD is relatively stronger on Nguyen and R* than on Livermore in our evaluation. One possible reason is that linear-style symbolic feature combinations are easier to exploit on those benchmarks, but are less well aligned with broader benchmark families such as Livermore.
>
> PySR is already explicitly reported in our SRBench results (Table 2), and Operon is already included in the SRBench evaluation and shown in Appendix C.6.1 / Figure 7. We will revise this part to make the benchmark coverage and comparability clearer.
>
> Q2&Q3.
> We agree that the depth-efficiency tradeoff is real. As explained above and in our response to Reviewer u9om Q1, the core value of SPBT+GCN is not reduced computational cost, but the ability to make strong global constraints based on ancestor-path information enforceable in batch-compatible generation. For scientific symbolic regression, these constraints help exclude expressions that may fit numerically in a local sense but are structurally less reasonable, which is why we view this tradeoff as worthwhile in our setting. We also do not rely solely on deeper GCN scaffolds: the appendix shows that D=4/5 already provides the best recovery-runtime tradeoff, while deeper scaffolds become substantially more expensive. This is why GCN is used to generate high-quality constrained fragments, and GP is used as a complementary explicit tree-space stage rather than forcing all expressivity into a deeper fixed scaffold.
>
> Q4.
> We thank the reviewer for pointing out this related work. We agree that recent neural SR methods such as GFN-SR already go beyond flat token sequences and adopt tree-aware generation. However, the key issue we focus on is not tree modeling in general, but whether strong global constraints can be enforced as hard rules during batch neural decoding. Many constraints of interest are not merely local validity conditions; they depend on ancestor-path or subtree-level context. Our main point is therefore not simply that SPBT represents trees, but that it provides a fixed-topology, batch-compatible interface in which whole-path or subtree-dependent masking becomes directly implementable during generation. We will clarify this added value of SPBT more explicitly in the revision; see also our response to Reviewer W3h3 for the broader methodological motivation.
>
> Q5.
> We agree that the discussion of limitations is currently insufficient. In the revision, we will add a dedicated subsection explicitly discussing three points: (1) the tradeoff between expressiveness and efficiency induced by the maximum SPBT depth, since shallow scaffolds limit the class of expressions that can be represented directly by the neural decoder, while deeper scaffolds introduce more redundant structure and higher runtime; (2) the computational overhead of whole-scaffold GCN decoding, because each decoding step still performs message passing on the full fixed tree; and (3) the remaining scalability challenges for more complex formulas and higher-dimensional settings, where symbolic regression in general faces a rapidly expanding combinatorial search space, and in our framework richer variable interactions may require larger candidate structures, increasing the pressure on the fixed scaffold, while also making the downstream refinement stage more demanding. We will clarify these limitations more explicitly in the revision.

---

> > ### Author Rebuttal · Reviewer_uRiL · 2026-04-03
> >
> > I acknowledge that I have read the author rebuttal. My concerns have been adequately addressed.

---

> > > ### Author Response · Authors · 2026-04-06
> > >
> > > Thank you very much for your careful review and constructive feedback. We sincerely appreciate your acknowledgement and are glad that our rebuttal has addressed your concerns.

---

### Official Review · Reviewer_u9om · 2026-03-07

**Soundness:** 3
**Presentation:** 3
**Significance:** 2
**Originality:** 2
**Overall Recommendation:** 4
**Confidence:** 3

**Summary:**

This paper proposes a graph-based symbolic regression framework (GCN-SR) that generates expressions as trees rather than linearized token sequences. The idea is to embed expression trees into a fixed-topology scaffold called Symbolic Perfect Binary Trees, thereby transforming symbolic expression generation into node attribute prediction on a fixed graph structure. This enables batched decoding and enforcement of global semantic constraints via precomputed ancestor tables. For training, the authors introduce Similarity-Weighted Policy Gradient, which leverages genetic programming only to construct similarity-weighted reward signals for on-policy samples rather than using GP-refined elites as training targets, thereby mitigating distribution mismatch. Experiments on Nguyen, Livermore, R*, and SRBENCH benchmarks demonstrate that GCN-SR achieves SOTA recovery rates compared to several neural and hybrid baselines.

**Compliance With Llm Reviewing Policy:**

Affirmed.

**Final Justification:**

The authors provided a thorough rebuttal that addressed most of my concerns. Overall, I raise my score to weak accept (4). The SPBT representation and the similarity-weighted policy gradient are original contributions. The remaining **scalability concern** prevents me from recommending a higher score. The current method appears effective in a **moderate-depth regime** but **its applicability beyond that regime is not yet clear**.

**Key Questions For Authors:**

1. In adopting LCS as the structural similarity measure, did the authors evaluate alternatives such as tree edit distance[^1] or exact subtree matching[^2]?

[^1]: Akutsu, Tatsuya, et al. "Tree edit distance with variables. measuring the similarity between mathematical formulas." _arXiv preprint arXiv:2105.04802_ (2021).

[^2]: Burlacu, Bogdan, et al. "Hash-based tree similarity and simplification in genetic programming for symbolic regression." _International conference on computer aided systems theory_. Cham: Springer International Publishing, 2019.

**Limitations:**

The paper does not include a dedicated "Limitations" or "Broader Impact" section. The authors present some limitations through their experiments and appendices. In Appendix C.5.2, the authors discuss the trade-off between SPBT tree depth, noting that a shallow depth limits expressive capacity while an excessive depth introduces redundant structure and significantly increases runtime. The authors could further discuss the scalability limitations of their approach when applied to higher-dimensional problems.

**Strengths And Weaknesses:**

**Strengths**:
1. SPBT maps variable-topology expression trees to a fixed-topology perfect binary tree and performs node-attribute prediction with a GCN. This enables aggregation of subtree and ancestor information and yields structure-aware decoding. This design reformulates tree generation as structured node-attribute prediction and supports batched parallel computation.
2. The paper analyzes the distribution mismatch induced by using GP elites as training targets and adopts an intermediate strategy. GP is used only to construct reward signals, and gradient computation is restricted to on-policy samples. The two propositions provide mathematical support for stable convergence. The proofs are extensible and are independent of the specific network architecture.

**Weaknesses**:

1. **Computational efficiency**: SPBT embeds variable-topology expression trees into a fixed perfect binary tree scaffold of depth $D$. This gives a unified graph structure and makes batching easier. While real expressions usually occupy only a small fraction of these positions, and the rest should be filled with `<empty>`. At every decoding step, the model should run a full GCN forward pass over the entire scaffold ($|V| = 2^D - 1$ nodes with $L$ message-passing layers), causing the compute cost to grow exponentially with $D$. Table 12 shows that when $D = 6$, each epoch takes about 1.35 seconds, which is about 11× slower than $D = 4$. The paper does not does it provide a complexity comparison against sequence-based decoders whose cost grows roughly linearly with sequence length. For complex scientific formulas, the scalability of the current approach is concerning.
2. **The choice of similarity function (LCS)**: LCS is a sequence-matching algorithm. Using it on preorder traversal sequences to measure structural similarity has limitations. It is insensitive to algebraic structure and cannot recognize basic mathematical equivalences. For example, under commutativity, $x + y$ and $y + x$ can be judged only partially similar rather than fully equivalent because their traversal orders differ; under associativity, $(a + b) + c$ and $a + (b + c)$ have different tree shapes. More generally, being close in a preorder sequence is not the same as being close in tree topology—two trees with very different structures may get a high LCS score simply because they share similar leaf-token subsequences, which can mislead the reward signal.
3. Experimental evidence suggests that GP is the dominant contributor. In the ablation study (Table 3), removing GP reduces recovery from 89.5% to 29.5%, whereas removing semantic constraints reduces recovery from 89.5% to 86.4%. On SRBENCH, GCN-SR recovers 61/100 Feynman equations, while PySR reaches 53/100. The gap is not large, and PySR does not require neural network training.

---

> ### Author Rebuttal · Authors · 2026-03-31
>
> We thank the reviewer for the careful reading and thoughtful comments. We address the questions below.
>
> Q1.
> We agree that the depth-efficiency tradeoff is real, and do not claim that SPBT-based decoding is computationally cheaper than sequence decoding. Each decoding step requires a full GCN forward pass on the fixed scaffold, so the cost scales with scaffold size and GCN depth; Appendix Tables 11–13 quantify this tradeoff.
>
> Our point is not that SPBT reduces computation, but that it provides an explicit tree interface where hard ancestor-path-dependent constraints become batch-enforceable. This is also why the framework is not designed to rely solely on deeper scaffolds. The overhead is not uniformly prohibitive: in Table 12, GCN(D=4) is faster than LSTM (0.1197 s/epoch vs. 0.2350), GCN(D=5) is only moderately slower (0.3328 vs. 0.2350), and the sharp slowdown mainly appears at D=6 (1.3521). Table 13 further shows that D=4 and D=5 already achieve 100% average recovery on Nguyen-1~11 with 29.98 s and 41.90 s average runtime, whereas D=6 increases runtime to 109.85 s without improving recovery. We therefore view the current method as targeting a moderate-depth regime: SPBT provides constrained neural proposals, and GP complements it as an explicit tree-space stage rather than forcing all expressivity into a deep fixed scaffold.
>
> Q2.
> We agree that LCS is not a perfect tree-aware similarity and cannot fully capture general algebraic equivalence. The main text already applies normalization/canonicalization, which reduces some superficial traversal-order mismatch, but this does not eliminate the limitation you pointed out. To check whether our conclusion depends strongly on LCS, we also evaluated two alternatives already implemented: unit-cost TED, and a simplified subtree-matching baseline that partitions both target and candidate expressions into subtrees of depth ( 2 <= d < 5) and computes normalized overlap between the resulting subtree sets. On Livermore, the results are: LCS 89.5 ± 28.0, TED 87.05 ± 27.2, and simplified subtree matching 81.82 ± 29.34, with LCS performing best in our current setting. Our claim is therefore not that LCS is the optimal tree similarity in general, but that a lightweight structure-guided reward is useful in this neural-to-GP feedback loop.
>
> Q3.
> We agree that GP is an important component of the full system. However, the large drop in the w/o-GP ablation (89.5% -> 29.5%) should not be interpreted as showing that the overall gain is explained by GP alone. In our setting, the neural generator operates on a fixed-depth SPBT scaffold. This design makes strong global constraints batch-enforceable, but also imposes a hard representational limit: if an exact target expression requires a tree deeper than the preset scaffold, then pure GCN decoding on that scaffold cannot represent the target tree, so exact recovery by pure GCN alone is impossible in such cases.
>
> A more direct view of the neural contribution is given by the comparison between Random GCN and the full GCN-SR. Here, Random GCN keeps the same downstream GP refinement and the same evaluation budget, but replaces the trained GCN generator with an untrained random network. Under this setting, recovery drops from 89.5% to 58.2%, showing that the learned constrained neural proposal itself contributes substantially. This is also consistent with the other ablations: even when GP is kept, weakening the neural-GP coupling still degrades performance (80.5% w/o reward refinement, 76.8% w/o similarity weighting, and 82.3% with PQT). Thus, the gain does not come from GP alone, but from how GP is coupled to a learnable structured generator.
>
> Regarding the comparison with PySR, we agree that the margin on SRBench is smaller than on our noiseless white-box benchmarks. We nevertheless believe this result remains meaningful because it is obtained on the Feynman benchmark under 0.1% relative noise, where reward-trained neural symbolic regression becomes more difficult: under noise, structurally incorrect expressions can still obtain competitive fitting scores, making the reward signal less effective at favoring the true symbolic structure. In this setting, GCN-SR recovers 61/100 equations, while PySR recovers 53/100, compared with 59/100 for PySR under noiseless data. A similar trend is reflected by uDSR, a neural+GP hybrid baseline, which drops from 72/100 under noiseless data to 40/100 under 0.1% relative noise, whereas GCN-SR decreases from 76/100 to 61/100. We therefore view this result as consistent with the benefit of coupling GP with a learnable structured neural generator under noise: the margin over a strong GP baseline becomes smaller, but is still preserved, while the degradation is substantially milder than in another neural+GP baseline.
>
> We also agree that higher-dimensional scalability deserves clearer discussion; please see our response to Reviewer uRiL Q5.

---

> > ### Author Rebuttal · Reviewer_u9om · 2026-04-03
> >
> > I thank the authors for their detailed responses.
> >
> > Q1: I acknowledge the authors' clarification that the core value of SPBT lies in constraint enforcement rather than computational savings. However, a remaining concern persists: for formulas that exceed $D=5$, does the neural component's contribution diminish relative to GP? A characterization of how the neural-vs-GP contribution ratio shifts as target complexity grows would further strengthen the argument.
> >
> > Q2: I appreciate the additional comparisons against TED and subtree matching. The results demonstrating that LCS outperforms these alternatives in the current setting are informative.
> >
> > Q3: The Random GCN ablation (58.2% vs 89.5%) is a valuable addition and demonstrates that the trained neural generator contributes beyond merely feeding random proposals to GP. This adequately addresses my concern.

---

> > > ### Author Response · Authors · 2026-04-04
> > >
> > > Thank you for this helpful follow-up. To better characterize how the learned GCN contributes as target complexity grows, we re-organized the existing Livermore ablation results by target-expression depth relative to the scaffold limit.
> > >
> > > Specifically, we group each Livermore task by the minimum binary-tree depth needed to represent the target expression under the same symbol library and equivalence handling used in the main paper. Since the key issue here is whether the target can still be directly represented by the fixed-depth (D=5) SPBT, we report the split relative to the scaffold limit: depth (≤5) (12 tasks) and depth (>5) (10 tasks). Under the same downstream GP budget, the grouped results are:
> > >
> > > | Model      | depth (≤5) (12 tasks) | depth (>5) (10 tasks) |
> > > | ---------- | --------------------: | --------------------: |
> > > | GCN-SR     |           84.2 ± 37.0 |            96.0 ± 8.4 |
> > > | w/o GP     |           54.2 ± 37.8 |             0.0 ± 0.0 |
> > > | Random GCN |           62.5 ± 47.5 |           53.0 ± 46.2 |
> > >
> > > Two observations are most relevant here. First, for depth (>5), the w/o-GP variant drops to 0.0, which is consistent with the representational limit of a fixed-depth (D=5) SPBT: in this regime, the neural generator alone cannot directly realize the full target expression. Second, under the same GP budget, full GCN-SR still clearly outperforms Random GCN in both groups, including depth (>5) (96.0 vs. 53.0). This suggests that even when the target exceeds the scaffold limit, the learned GCN still improves downstream GP by biasing the search toward better-structured proposals.
> > >
> > > Importantly, we do not view these numbers as an exact additive decomposition of “neural contribution” and “GP contribution,” since the two components interact. Rather, they provide a useful characterization of how the contribution shifts with complexity. Within the scaffold limit, the learned GCN improves over Random GCN by 21.7 points (84.2 vs. 62.5) and also retains a 30.0-point stand-alone exact-recovery contribution over w/o GP (84.2 vs. 54.2). Beyond the scaffold limit, stand-alone exact recovery by the neural generator necessarily vanishes in the w/o-GP setting (0.0), but under the same GP budget the learned GCN still provides a substantial 43.0-point gain over Random GCN (96.0 vs. 53.0). In this sense, as target complexity exceeds the scaffold limit, the learned GCN’s contribution shifts away from stand-alone exact recovery and toward search guidance for GP refinement, rather than disappearing.
> > >
> > > We also note that this split is intended to characterize representability relative to the fixed scaffold, not to define a monotonic difficulty ordering over Livermore tasks; the absolute averages also depend on the composition of tasks in each group. We will incorporate this grouped analysis and corresponding clarification into the revised manuscript.

---

### Official Review · Reviewer_W3h3 · 2026-03-13

**Soundness:** 2
**Presentation:** 3
**Significance:** 2
**Originality:** 2
**Overall Recommendation:** 4
**Confidence:** 4

**Summary:**

In this paper, we propose GCN-SR: We embed an expression tree of variable topology into a fixed topology via SPBT, and then use GCN to decode the tree structure with global constraints, thus preserving the hierarchy and imposing semantic constraints more naturally.
Meanwhile, the authors design SWPG, which incorporates the local optimization ability of GP into policy gradient training with strict on-policy in the form of similarity weighted rewards, and achieves better recovery rates than existing methods on multiple symbolic regression benchmarks.

**Compliance With Llm Reviewing Policy:**

Affirmed.

**Final Justification:**

My problem was solved and I decided to raise my score.

**Key Questions For Authors:**

1. What is the difference between SPBT specifying a maximum depth and autoregression specifying a maximum length? What are the advantages?

2. What is the difference between constraint-aware decoding mentioned in the paper and regular constraints?

3. This paper will calculate the structural similarity between the original sample and elite as part of the reward. What is the motivation for this? How do you calculate their similarity? Does a high similarity to elite necessarily mean a high R2? Does this cause the model to be misleading?

**Limitations:**

1. SPBT relies on a preset maximum depth, which requires a manual tradeoff between expressivity and efficiency.
2. GCN has to do message passing on the whole fixed tree at each step, and the overhead will increase significantly when the depth goes up;

**Strengths And Weaknesses:**

**Strengths**

1. The algorithm innovatively calculates the structural similarity between the original sample and the elite as a part of the reward, and verifies a good effect.

2. The algorithm improves the algorithm based on GP. The experimental results show that the improved algorithm improves the performance of the algorithm.

**Weaknesses**

1. The algorithm is a modification of existing methods (especially similar to the overall process of NGGP), and the real innovation is limited.

2. How is the way this algorithm adds constraints different from existing algorithms, such as those in DSR?

3. The experimental dataset in this paper is not comprehensive enough. Please test other datasets, Constant, Jin, Neat, Keijzer, etc

4. In this paper, the structural similarity of the original on-policy sample and elite is used to allocate rewards, which is innovative, but I am worried that it is also a disadvantage, because in symbolic regression, even though the expression sequence is very similar, its final evaluation index R2 may be far different. So I doubt the rationality of this approach.

---

> ### Author Rebuttal · Authors · 2026-03-31
>
> We thank the reviewer for the feedback. We agree that the current introduction could better highlight the main novelty, which may make the work appear closer to a neural+GP hybrid or an NGGP-style variant. We will clarify this distinction more explicitly in the revision.
>
> Our contribution is not merely a neural+GP loop, and the difference from NGGP-style hybrids is twofold:
>  (1) SPBT+GCN changes the decoding interface by providing a batch-compatible tree state that preserves hierarchy and makes ancestor-path-dependent hard constraints enforceable during neural generation; and
>  (2) SWPG changes the feedback mechanism by feeding GP-discovered improvement back through reward shaping, rather than directly matching off-policy GP elites.
>
> Many neural SR methods linearize an expression tree into a token sequence. This is convenient for batched autoregressive generation, but the hierarchy is not explicitly maintained during decoding. As a result, such methods more naturally support local validity constraints, but not hard global constraints requiring ancestor-path checking. For example, local constraints usually inspect immediate parent-child relations, whereas stronger global constraints may require checking the full ancestor path, e.g., limiting repeated operator nesting or restricting whether f(x) in  sin(x+f(x)) may itself contain sin. Such global constraints help preserve interpretability and control structural complexity, which can also reduce overfitting risk.
>
> SPBT+GCN is therefore not just “adding more rules.” Its role is to provide a batch-compatible structural interface that preserves hierarchy and exposes whole-path information, making hard ancestor-dependent constraint checking implementable during decoding. Empirically, the appendix already shows that, under a matched evaluation budget on Nguyen, GCN+SPBT with global constraints improves average recovery from 48.33% / 51.67% to 57.50%, while reducing the evaluation budget on successful runs from 597.22k / 903.74k to 423.11k (Appendix C.1).
>
> We acknowledge this tradeoff: SPBT requires a predefined maximum depth, and GCN performs message passing on a fixed scaffold. Thus, for deeper or more complex formulas, fixed-depth constrained neural decoding may not directly generate the full expression, but it can still provide high-quality constrained fragments. We thus combine it with GP refinement. GP is introduced not to repeat an existing neural+GP paradigm, but because it is well suited to explicit tree-space search, where crossover and mutation can reorganize constrained fragments into more complete expressions. SWPG then feeds the improvement revealed by GP refinement back to the generator through a similarity-weighted reward, rather than directly using off-policy GP elites as supervision. We will make this distinction clearer in the revision.
>
> Q1.
> The maximum length in autoregression limits only the number of generated tokens; it does not explicitly preserve tree hierarchy and therefore does not naturally support ancestor-path-based constraint checking. In contrast, the maximum depth in SPBT defines a persistent tree state. This makes whole-path constraints such as nesting limits or ancestor-dependent operator restrictions naturally enforceable during decoding.
>
> Q2.
> In DSR-like sequence generation, constraints are typically local validity rules, because they depend mainly on the current symbol and its immediate parent-child relations. By contrast, our constraint-aware decoding targets stronger global structural constraints that require persistent ancestor-path checking. The distinction is therefore not merely “more constraints,” but a different decoding interface: SPBT maintains an explicit tree state and supports batched ancestor queries, making hard whole-path constraint checking implementable during generation.
>
> Q3.
> We agree that high structural similarity does not imply high R^2, and we do not use similarity as the final metric or as a symbolic-equivalence criterion. It is only an auxiliary reward term; optimization is still driven by fitting quality. Its role is to provide more informative credit assignment between the current neural proposal and the GP-refined elite, not to replace the identification of high-quality expressions. In the current implementation, similarity is computed on normalized preorder representations via LCS. We also evaluated alternative structure-aware measures, including TED and a simplified subtree-matching baseline; please see our response to Reviewer u9om Q2.
>
> Other comments.
>
> Regarding dataset coverage, we agree that the original submission could be broader. We have evaluated the Jin constant benchmark in Appendix C.5.3 (page 22), and we now also report Neat and Keijzer here: Neat: 83.33±33.91; Keijzer: 75.33±39.97. Although some expressions in Neat/Keijzer overlap with Nguyen/Livermore, we agree that reporting them makes the empirical picture more complete; we will include them in the revision.

---

> > ### Author Rebuttal · Reviewer_W3h3 · 2026-04-04
> >
> > Dear author, thank you for your reply. I have the following questions. If you can reply, I will agree to raise the score.
> >
> > What are the advantages of SPBT over the DSR or DSO method of checking whether generation has stopped (checking whether a complete expression tree has been generated) : count = count + ararity - 1 to impose constraints? The way count= count + arity-1 works is that count is initialized to 1, and arity is the primitive number of symbols. If the node count=0, then a complete expression rooted at that node is generated. On the other hand, if the count is not equal to 0, then the subtree is generated for that node, and we just mask out the symbols we want to constrain during the generation process. I think it's more elegant to add constraints on the fly during the generation process, and it can also impose global constraints instead of local constraints as you said. For example: for [+,x,log,*,x,x]. Now that I have the symbol log, which triggers the constraint, I just set the log node's count_log=1, and as long as count_log is not equal to 0, I mask out any symbols that violate the constraint (e.g.,sin). This can also implement global constraints, and is quite elegant. So I would love to know what are the advantages of SPBT, I feel that SPBT is even more cumbersome?

---

> > > ### Author Response · Authors · 2026-04-05
> > >
> > > Thank you for the helpful follow-up. We agree that sequence-based decoding can enforce ancestor-dependent constraints, and that for a simple scoped rule such as your count_log example, sequence-side scope tracking can be a valid and lightweight solution. Our intended distinction is therefore not expressivity in principle, but the representation interface: in sequence decoding, the relevant structural scope is recovered online from the prefix history, whereas in SPBT it is maintained explicitly as part of the decoder state.
> > >
> > > This distinction becomes practically relevant when exact structure-dependent constraints must be implemented and extended within a batched neural decoder. For a particular scoped rule, a sequence decoder can indeed introduce dedicated states such as count_log. However, as the constraint family expands to multiple ancestor types, simultaneously active triggers, nested scopes, or sibling/subtree-related conditions, sequence-side enforcement typically requires additional rule-specific parser states and bookkeeping maintained over each sample’s prefix history.
> > >
> > > In SPBT, by contrast, these cases are expressed through the same explicit structural interface built on the current node, its indexed ancestors, and scaffold-defined relations. New structural rules can therefore be specified over the same explicit representation, rather than translated into additional online bookkeeping logic. This also matters in batched training and sampling: under SPBT, node positions and ancestor indices are aligned across the batch, enabling shared tensor gather/masking operations, whereas sequence-side enforcement typically depends on per-sample prefix states, so achieving the same kind of batch-aligned masking is generally less direct and may require additional implementation-specific design under standard vectorized batching.
> > >
> > > Beyond this interface-level difference, the SPBT-based design also shows an empirical evaluation-budget advantage when paired with GCN. Under local-only constraints, GCN+SPBT achieves recovery comparable to the sequence model while requiring substantially fewer evaluated candidate expressions on successful runs (48.33% vs. 51.67% recovery with 597.22k vs. 903.74k evaluation budget). We also added a supplementary comparison in which the sequence baseline is equipped with analogous global constraints through sequence-side scoped-state tracking. Under this setting, GCN+SPBT still achieves slightly higher recovery with a lower evaluation budget (57.5% vs. 55.8% recovery with 423.11k vs. 746.43k budget). This empirical pattern suggests that the explicit tree representation in SPBT may provide a stronger structural basis for GCN-based decoding, helping the model better exploit hierarchical dependencies during search and thereby achieve similar or better recovery with a lower evaluation budget.
> > >
> > > In summary, the practical advantage of SPBT in our framework is that it makes structural scope explicit and batch-aligned in the decoder state. This allows ancestor/subtree-aware hard masking to be implemented more naturally within standard batched neural decoding, and also makes the framework easier to extend in practice, since new structural rules can be added on the same explicit representation rather than encoded as separate rule-specific prefix state machines. We will revise the manuscript accordingly to make this point clearer.

---

### Official Review · Reviewer_VNMZ · 2026-03-20

**Soundness:** 3
**Presentation:** 3
**Significance:** 3
**Originality:** 3
**Overall Recommendation:** 4
**Confidence:** 4

**Summary:**

This paper introduces GCN-SR, a graph-based symbolic regression that generates expressions directly in tree form as opposed to linearized sequences. This enables batched decoding of expressions with arbitrary topologies, while explicitly retaining the full hierarchy so that global semantic constraints can be enforced as hard rules during generation. Symbolic regression systems often benefit from a closed loop system, whereby a neural generator can be improved by considering genetic programming-improved expressions. The authors introduce similarity-weighted policy gradient to train the generator, by leveraging the genetic program's output to construct rewards, allowing the neural generator to benefit from evolutionary refinement without directly training on GP-refined expressions as targets, avoiding a distribution mismatch.

**Compliance With Llm Reviewing Policy:**

Affirmed.

**Final Justification:**

I maintain that the paper offers a simple yet effective contribution. I do have my gripes about the presentation as listed above, which is why I will stick with my weak accept.

**Key Questions For Authors:**

Please see my questions in the above "Weaknesses" section.

**Limitations:**

I would appreciate if the authors include a limitations section.

**Strengths And Weaknesses:**

Strengths:

- I found the paper to be largely well written, with the exposition supported by illustrative figures.

- The proposed approach is straightforward yet quite effective as the authors demonstrated empirically. I do believe the proposed
approach is likely to be widely adopted

Weaknesses:

- I found Figure 1 to be needlessly confusing: It is my understanding that in Figure 1A the GNN act as scoring functions for GP/MCTS Search, and therefore, 1A is actually a closed loop where the GP/MCTS search returns an expression tree that is scored by the GNN. That scores is then utilized by the GP/MCTS search somehow to reach a better expression tree? Furthermore, I believe there should be a GP/MTCS box in Figure 1B which loops back from the generated expression tree to the GCN

- It wasn't immediately clear to me what *on-policy* refers to on line 133; I believe it refers to the GCN? I do not believe there had been any prior discussion of a policy (and I am not sure that such terminology is justified here?)

- Relatedly, the paragraph starting line 155 (right) on (SWPG) could use a gentler introduction to policy/policy-gradients.

- It is not clear to me what SPBT-SLs are, why they are needed, and what the authors mean when they say "To preserve each node's identity [...] we augment SPBT with self-loops"

- How necessary is it to enforce the constraints in the constraint-aware decoding section? i.e. does omitting them lead to a drastic drop in performance?

- In lines 298-302 (left), the authors state that "direct mixing is biased" but then they mention that "off-policy terms can dominate the update and amplify variance". I think both statements need some support as it's not immediately clear where the bias and variance would seep in from. Relatedly, it is not clear to me that the surrogate objective is unbiased, or that it necessarily has lower variance (it is my understanding that C can be any positive number?)

- I find it very interesting (and maybe a bit concerning, or confusing) that the performance of GCN-R is more or less (sometimes exactly!) the same across $N \in \{\20, 100, 1000\}$, whereas the performance of other methods seems to depend greatly on the dataset size. Do you have any explanation or intuition for why that is the case? I am alarmed by how large the stds are. Could you explain why they're so large?

- Are the runtimes for GCN-SR comparable to the baselines?

---

> ### Author Rebuttal · Authors · 2026-03-31
>
> We thank the reviewer for the careful reading and positive assessment. Below we address the points.
>
> Q1.
> We agree that Figure 1 is not clear enough. In particular, it does not clearly distinguish prior GNN-guided search, where the GNN mainly serves as a scorer or guide for GP/MCTS, from our setting, where the GCN is the generator itself and GP is used for refinement and reward construction. We will redraw Figure 1 to make this distinction and the closed loop clearer.
>
> Q2.
> In our paper, “on-policy” refers to the current generation distribution parameterized by the GCN: expressions are sampled from the current generator, and the gradient is taken with respect to that same generator. We use this term to emphasize that GP-refined expressions are not treated as direct supervision targets; they affect learning only through reward construction. We agree that this terminology was introduced too abruptly and will clarify it in the revision.
>
> Q3.
> SPBT-SL denotes the message-passing graph obtained by adding a self-loop to every node in SPBT (Appendix C.2.1, page 17). Without self-loops, a node would aggregate only neighboring information; self-loops preserve its structural identity while still allowing contextual aggregation. An additional post-submission Nguyen check is consistent with this design choice, showing a drop from 57.50% to 54.38% after removing self-loops.
>
> Q4.
> These constraints are important rather than optional heuristics. Appendix C.1 (page 16) already shows that, under a matched Nguyen evaluation budget, GCN+SPBT with global constraints improves average recovery from 48.33% / 51.67% to 57.50%, while reducing the successful-run evaluation budget from 597.22k / 903.74k to 423.11k. More broadly, such constraints are hard to impose in standard sequence decoding but become directly enforceable once hierarchy and ancestor-path information are preserved.
>
> Q5.
> We agree that the wording in the current draft is too strong. If GP-refined expressions are directly mixed into the update, they are not sampled from the current generation distribution and do not come with a tractable, explicitly modeled sampling probability for rigorous correction. Such an update therefore cannot be interpreted as a standard on-policy score-function estimator. The appendix instead analyzes a stop-gradient surrogate objective with the elite set fixed. Under that surrogate, the same-batch baseline introduces only a small bias of order O(1/B), and the variance discussion should be understood as a boundedness/scaling statement under bounded assumptions, rather than a blanket claim that the method is always lower-variance.
>
> Q6.
> We agree that this point deserves a clearer explanation. We do not interpret the flat trend as meaning that GCN-SR is generally insensitive to dataset size. In our setting, performance is measured by exact symbolic recovery on noiseless white-box benchmarks, with a newly sampled dataset for each run. Under this protocol, once a task is already sufficiently identified at N=20, increasing N further may not materially change the recovery outcome. This is consistent with Table 1, where GCN-SR is already close to saturation across N: 93.3/93.3/93.3 on Nguyen, 90.0/90.5/91.8 on Livermore, and 100/100/100 on R*. Relative to other methods, we interpret this as GCN-SR reaching this regime earlier because the explicit tree scaffold and global constraints reduce structurally implausible candidates. Thus, once a modest number of samples is sufficient to distinguish the target from most competing structures, the remaining failures are more likely to come from search/representation bottlenecks than from finite-sample estimation noise. We agree that our main-text wording on the standard deviations was imprecise and will revise it.
>
> Regarding the large standard deviations, the numbers in Table 1 are sample standard deviations across tasks within each benchmark, rather than instability on a single task. Since task difficulty varies substantially, and many task-level recovery rates are close to either 0% or 100%, the across-task standard deviations can remain large even when behavior within each task is relatively stable.
>
> Q7.
> We agree that runtime comparisons should be interpreted carefully. Runtime comparisons between GCN and LSTM are already reported in Appendix Tables 11–13, where both models are implemented in the same framework and allow a controlled matched-capacity comparison. As supplementary post-submission evidence obtained under the same computational resources on Nguyen (1–11), DSR, NGGP, and RSRM require 172.33, 63.76, and 76.59 seconds, respectively, while GCN-SR requires 41.90 seconds. The gain is not from cheaper per-step decoding, but appears to come mainly from reducing the number of expressions that need to be evaluated, consistent with Appendix C.1 / Table 5.
>
> Q8.
> We agree and will add a dedicated limitations section; please also see our response to Reviewer uRiL Q5 for a more detailed discussion of this point.

---

> > ### Author Rebuttal · Reviewer_VNMZ · 2026-04-04
> >
> > Thanks for taking the time to respond to my concerns. Almost of my concerns not pertaining to presentation have been addressed.

---

> > > ### Author Response · Authors · 2026-04-06
> > >
> > > Thank you very much for your careful review and constructive feedback. We sincerely appreciate your acknowledgement and are glad that our rebuttal has addressed your concerns.

---

### Decision · Program_Chairs · 2026-04-30

**Decision:**

Accept (regular)

**Comment:**

All four reviewers acknowledged the author rebuttals and two of them raised the scores.
As a result, this paper received four WAs.

Reviewers valued the effectiveness of the proposed method and its technical contribution, specifically about SPBT (Symbolic Perfect Binary Trees) proposed in this work.
SPBT triggered many discussions in this forum e.g., its design, objectives, computational efficiency, representation, and effectiveness. I find the overall sentiment on the proposed method positive.

Some of the reviewers suggested adding more baselines for comparison, which I think is not required for camera-ready, but should be considered in future studies.

My biggest concern at this stage is that the current revision does not provide hyperparameter tuning process or search space for **their baseline methods** while Appendix C.3.1 suggests wide ranges of hyperparameter values to tune for the proposed method.

I also have a concern that most of the experimental results and analysis in this work are based on relatively simple SR problems such as Nguyen, Livermore, JIN, Neat, and Keijzer. SRBench was used a few places in the paper, but the benchmark also has many issues as pointed out in [the SRSD benchmark](https://data.mlr.press/assets/pdf/v01-3.pdf).

That being said, overall I am leaning towards acceptance, but only if the there is room in the program to accept this work.